# Predicting scale-dependent chromatin polymer properties from systematic coarse-graining

Sangram Kadam [1] ✉, Kiran Kumari[1], Vinoth Manivannan [1], Shuvadip Dutta [2], Mithun K. Mitra [2] & Ranjith Padinhateeri [1,3] ✉

Simulating chromatin is crucial for predicting genome organization and dynamics. Although coarse-grained bead-spring polymer models are commonly used to describe chromatin, the relevant bead dimensions, elastic properties, and the nature of inter-bead potentials are unknown. Using nucleosome-resolution contact probability (Micro-C) data, we systematically coarse-grain chromatin and predict quantities essential for polymer representation of chromatin. We compute size distributions of chromatin beads for different coarse-graining scales, quantify fluctuations and distributions of bond lengths between neighboring regions, and derive effective spring constant values. Unlike the prevalent notion, our findings argue that coarse-grained chromatin beads must be considered as soft particles that can overlap, and we derive an effective inter-bead soft potential and quantify an overlap parameter. We also compute angle distributions giving insights into intrinsic folding and local bendability of chromatin. While the nucleosome-linker DNA bond angle naturally emerges from our work, we show two populations of local structural states. The bead sizes, bond lengths, and bond angles show different mean behavior at Topologically Associating Domain (TAD) boundaries and TAD interiors. We integrate our findings into a coarse-grained polymer model and provide quantitative estimates of all model parameters, which can serve as a foundational basis for all future coarse-grained chromatin simulations.

The eukaryotic genome is organized in the form of many long chromatin polymer chains, each essentially a string of nucleosomes—DNA wrapped around histone proteins—folded, looped, and condensed into domains of different compaction[1–4]. The spatial and temporal organization of these chains and their internal epigenetic states are crucial in deciding aspects ranging from cellular function to differentiation and development[5–8].

Chromosomes are typically simulated and studied as coarse-grained(CG) bead-spring polymer chains[9,10]. A coarse-grained polymer picture is useful for many reasons: it is nearly impossible to simulate the huge polymer set (~millions of nucleosomes) in its entirety. More importantly, the coarse-grained representation, with effective parameters, can be a powerful tool to understand chromatin organization and dynamics and make useful predictions[11–24]. However, since we still do not understand the chromatin structure and properties in detail, systematic coarse-graining has been a difficult task. We do not fully know the polymer properties/parameters relevant for simulating coarse-grained chromatin.

[1]Department of Biosciences and Bioengineering, Indian Institute of Technology Bombay, Mumbai 400076, India. [2]Department of Physics, Indian Institute of Technology Bombay, Mumbai 400076, India. [3]Sunita Sanghi Centre of Aging and Neurodegenerative Diseases, Indian Institute of Technology Bombay, Mumbai 400076, India. ✉e-mail: sangramkadam@iitb.ac.in; ranjithp@iitb.ac.in

Owing to a large body of work over the past few decades, double-stranded DNA has a good coarse-grained description as a semi-flexible polymer[25–27]. We understand its coarse-graining size, bending stiffness, stretching elasticity, and other relevant parameters[26,27]. However, chromatin is a more complex polymer having heterogeneous properties arising from different epigenetic states, amount of different proteins bound, and local folding[28,29]. This complexity makes it difficult to accurately compute coarse-grained bead diameter, elastic constants, and other physical properties for a chromatin polymer.

Recent experimental advances have made it possible to understand chromatin structure using biochemical methods like Hi-C, Micro-C[30–41] and imaging methods like SAX, cryo-EM, and super-resolution imaging[42–49]. The studies so far show that chromatin is organized into different compartments and topologically associated domains (TADs)[2,7,31–33]. While histone modifications, transcription factors, and chromatin binding proteins greatly affect chromatin folding and make it a highly heterogeneous polymer, how the interplay between these factors decides the compaction and dynamics of chromatin is currently being investigated.

While different experimental methods provided us data to understand chromatin organization[30–33,35–37,42–44,46–48,50,51], theoretical/computational studies have been pivotal in understanding and explaining chromatin characteristics[11–15,18–22,24,52–67]. Models that simulated chromatin at nucleosome resolution primarily investigated how different molecular interactions influenced higher-order organization beyond the 10 nm chromatin[53,55,56,61,66,68–70]. Nearly all models that are employed to understand Hi-C/microscopy data represented chromatin as a bead-spring polymer chain with each bead representing chromatin of length in the range ~1 kb to 1Mb[12–17,20–24]. However, physical dimensions and elastic properties of chromatin are not well understood. What is the diameter of a 1 kb, 10 kb, or 100 kb chromatin bead? What is the magnitude of the spring constant that would represent the thermal fluctuation of chromatin at different length scales? Should chromatin be considered a flexible polymer or a semi-flexible polymer? Does it have an intrinsic curvature/bending stiffness? None of these questions have clear, definitive answers in the literature, currently. While it is well known that chromatin behavior is heterogeneous, depending on the epigenetic state, existing coarse-grained models of chromatin typically assume that the physical dimensions of the beads and elastic properties of the filament are uniform along the polymer, independent of the epigenetic state. How these properties—the size of the beads, stretching elasticity, bending elasticity, etc.—vary along the contour is also unknown. In the current models, the heterogeneity of chromatin is often incorporated into additional intra-chromatin interactions—interaction between two far-away beads[12,24,71]. Since one does not know the size of a coarse-grained chromatin bead, it is taken as a fitting parameter—a constant number across the filament—to achieve experimentally measured 3D distance values[12,15,22,24,72,73]. Supplementary Fig. 1 shows some of the reported values and how scattered they are. We do not understand this variability and what each number means.

One could not do systematic coarse-graining so far because the chromatin conformation capture data was available only with lower resolutions like 100 kb, 10 kb, and up to 1 kb[30–33,50]. Obtaining information smaller than the HiC resolution was not possible. Moreover, even at the smallest size scale (~kb), the physical dimension of chromatin—chromatin bead diameter—was an unknown parameter. However, recent experiments have provided us chromatin conformation capture data at near-nucleosome resolution—200 bp resolution, which is essentially a nucleosome plus the linker DNA[34,35,39] (also see Supplementary Fig. 2a). This data enables us to make a fine-grained chromatin model and systematically probe the properties of the coarse-grained chromatin. The advantage here is that the physical size of a 200 bp chromatin is not a completely unknown free parameter; we do have a fair idea about the size of a 200 bp chromatin bead. In this work, we take advantage of this recent fine-grained data, start from the 200 bp Micro-C contact map, and construct chromatin polymer configurations that satisfy the map. From our work, without any arbitrary fitting parameter, the 3D distances and radius of gyration values emerge in a reasonable range comparable to known experiments. Using the 200 bp-chromatin as a fine-grained polymer, we coarse-grain the chromatin systematically. This enables us to predict several quantities essential for anyone simulating a coarse-grained chromatin polymer. We predict (i) the physical sizes of coarse-grained beads of various chromatin length scales, (ii) the overlap between coarse-grained beads and an effective inter-bead soft potential energy, (iii) the value of the spring constant between neighboring beads dictating the fluctuation, and (iv) the distributions of bond angles and dihedral angles giving insights into the stiffness of chromatin. We show that some of the ideas we learned—e.g., soft inter-bead interactions that allow overlap—are crucial for obtaining sensible 3D distances/$R_g$ when coarse-grained models are employed.

## Results

### Constructing 3D chromatin configurations at near-nucleosome resolution consistent with Micro-C contact map and measured 3D distances

We simulated a fine-grained chromatin polymer made of "nucleosome-linker" (NL) beads, with each bead representing 200bp of chromatin (Fig. 1a, Methods and Supplementary Information (SI)), and generated an ensemble of steady-state chromatin configurations, taking the Micro-C contact probability data ($P_{ij}$) of mouse embryonic stem cells (mESCs) as input[35]. We simulated ten different genomic loci (see Supplementary Table 1) having broad euchromatic or heterochromatic chromatin state characteristics. We computed the ensemble-averaged contact probability for each locus and compared them with the input contact map. The contact maps from the simulations appear visually similar to the Micro-C data (Fig. 1b–d and Supplementary Fig. 3). The bottom panel shows representative snapshots from the simulations. For the Ppm1g locus, the beads are colored based on the domains in the contact map (color-strip at the top of Fig. 1b–d), while for the Gm29683 and Cbx8 loci, the far-away heterochromatic regions interacting with each other are shown in red and blue color (Fig. 1c, d). Even though these are representative snapshots, one can see signatures of domain separation (configurations below Fig. 1b) and interaction among far-away regions (configurations below Fig. 1c, d). The contact probability versus genomic distance plots from simulations and experiments are comparable (Supplementary Fig. 4). We quantified the similarity between the experimental and simulation contact matrix by computing the stratum-adjusted correlation coefficient (SCC)[74] (see Supplementary Note 1D). The SCC values for most regions are above 0.9, suggesting that the simulations reproduce contact maps well (Supplementary Table 1). Beyond the contact map, we also compared the mean 3D distance from our simulations for the alpha globin region with the available experimental data[12,75] (see Fig. 1e). These results suggest that our simulations have generated an ensemble of configurations with the contact map and 3D distances comparable with experiments.

We now systematically coarse-grain all the above chromatin polymers. We chose $n_b$ consecutive NL beads to form a coarse-grained CG bead (colored big bead in Fig. 2a top right). The coarse-grained polymer consists of $N/n_b$ number of CG beads. We then measured various properties of the coarse-grained polymer such as the size of a CG bead $R_g$, bond length $l_{cg}$, bond angle $\theta_{cg}$ (Fig. 2a bottom), and dihedral angle $\phi_{cg}$ (see below). We study how these properties depend on the coarse-graining size $n_b$ and the genomic location. As a control, we have compared our chromatin results with the ideal chain (bead spring polymer with no self-avoiding interaction), the SAW (Self Avoiding Walk; bead spring polymer with Weeks–Chandler–Anderson

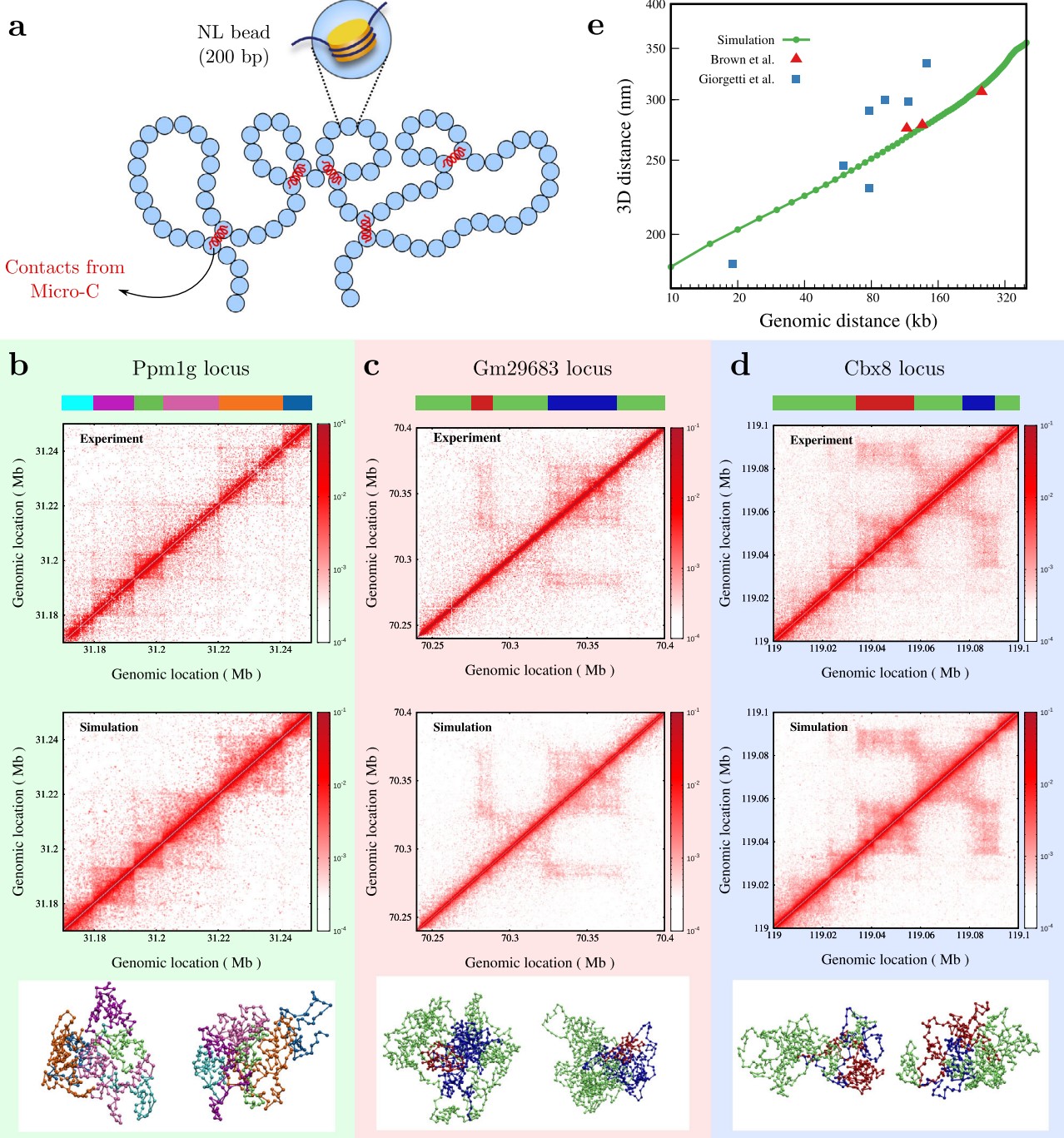

**Fig. 1 | Chromatin configurations are consistent with experiments. a** Schematic of the fine-grained chromatin polymer with one bead representing 200 bp nucleosome+linker (NL) chromatin. An ensemble of configurations is generated such that any pair of beads $(i, j)$ is connected (red springs) with a probability $p_{ij}$, based on observed contacts in Micro-C experiments by Hsieh et al.[35] (see text). **b–d** Comparison of contact maps obtained from our simulation to experiments for a euchromatic region (**b** Ppm1g locus) and two heterochromatic regions (**c** Gm29683 locus, and **d** Cbx8 locus). The bottom panel shows representative snapshots from the simulations, where the bead colors represent different domains, as shown in the color strip at the top. **e** 3D distance from our alpha globin simulation (green filled circles) compared with available experimental data for the same region (red filled triangles) taken from Brown et al.[75], and other regions (blue squares) of similar genomic length range taken from Giorgetti et al.[12]. Source data are provided as a Source Data file.

potential), and a highly packed globule (bead spring polymer with attractive Lennard-Jones potential with $\epsilon = 1k_BT$).

## Predicting the size of coarse-grained chromatin beads and its variability along the genome

Since nearly all polymer simulations use coarse-grained beads of various genomic lengths like 1 kb, 10 kb, 100 kb, etc., it is important to understand the physical size (radius) of such a coarse-grained bead. Does bead size depend on the state of the chromatin (heterochromatin or euchromatin) and/or the genomic location of the bead (TAD interior, TAD boundaries)? To answer this, we first computed the radius of gyration ($R_g$) of the $n_b$ consecutive beads that form a CG bead. Taking a sliding window of $n_b$ beads, we plotted the average radius of gyration ($R_g^i$) as a function of genomic location (Fig. 2b and Supplementary

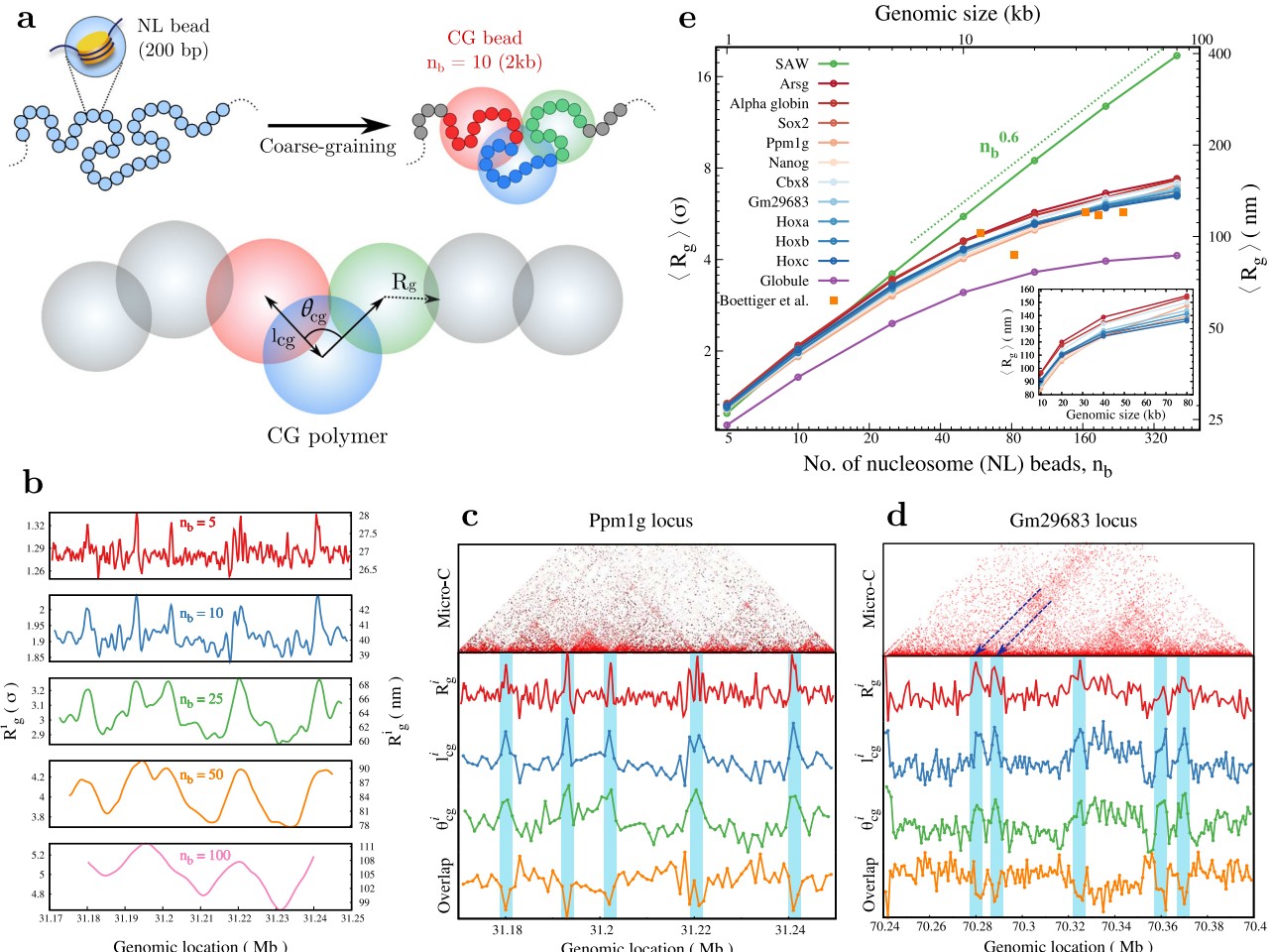

**Fig. 2 | Coarse-graining, bead size and its variability along the genome.**
**a** Schematic showing the coarse-graining procedure and quantities of interest. The fine-grained polymer of $N$ beads (small beads, representing 200 bp of nucleosome +linker (NL)) is coarse-grained into $N_{cg}$ big beads, with each big coarse-grained bead containing $n_b = N/N_{cg}$ beads. For illustration purposes, ten small colored beads ($n_b = 10$) are coarse-grained into one big CG bead. $l_{cg}$ is the length of the bond connecting two successive coarse-grained beads, $\theta_{cg}$ is the angle between two successive bonds, and $R_g$ is the radius of gyration of the coarse-grained polymer segment of size $n_b$. **b** Average radius of gyration at different genomic locations ($R_g^i$) for different $n_b$ values for the Ppm1g locus. **c**−**d** Variation of $R_g^i$ and other quantities ($l_{cg}^i$, $\theta_{cg}^i$, and overlap, see text) along the chromatin contour ($n_b = 5$) for (**c**) a euchromatic Ppm1g locus and (**d**) a heterochromatic Gm29683 locus. Note different behavior at the domain boundaries. **e** Mean $R_g$ for different coarse-graining sizes ($n_b$) showing the predicted range of values for different chromatin states. The chromatin loci with broad euchromatic characteristics are plotted in shades of red, while the loci with broad heterochromatic marks are plotted using shades of blue. Experimental data from Boettiger et al.[47] is added (orange squares; repressed chromatin domains in *Drosophila* cells) to demonstrate that the predicted $R_g$ values are reasonable. The dotted line is shown as a guide to the eye. $R_g$ values are presented in the units of NL bead size $\sigma$ ($Y_1$ axis) and also in nm ($Y_2$ axis). Source data are provided as a Source Data file.

Fig. 5a; real units in $X_2$ and $Y_2$ axes). For both euchromatic and heterochromatic segments, $R_g$ values vary along the genomic length for the same coarse-graining, representing heterogeneities along the chromatin. In contrast, $R_g$ curves for SAW and packed globule do not vary along the polymer contour (Supplementary Fig. 5b, c). To understand how the locations of the peaks and troughs in the $R_g$ correlate with the contact map, we compared them for a 1 kb coarse-graining scale (Fig. 2c, d and Supplementary Fig. 6). $R_g$ values peak at the boundaries of TAD-like domains and are relatively small within the interior of the domain. That is, coarse-grained beads representing inter-TAD regions will have a larger physical dimension. This is consistent with what is observed in recent experiments[42].

How big is a typical 1 kb, 5 kb, or 10 kb chromatin? We compared the mean $R_g$ values (averaged over the entire region we considered) for several genomic segments – some of them are euchromatic, and others are heterochromatic regions in terms of dominating histone modification marks (Fig. 2e). Throughout this paper, the chromatin loci with broad euchromatic characteristics are plotted in shades of red, while the loci with broad

heterochromatic marks are plotted using shades of blue. SAW polymer data is presented as a "control" having an expected $R_g \sim n_b^{0.6}$. Interestingly, even though there is variability among the gene region, they more or less fall in a narrow range. The range of average $R_g$ values for 1 kb, 10 kb, and 80 kb chromatin regions are 27–28 nm, 85–97 nm, and 135–155 nm, respectively (see Fig. 2e inset). The curves for SAW and globule mark the extreme values possible. The $R_g$ predictions for various regions from our simulations are of the same order as what is seen in experiments for the repressed chromatin domains in *Drosophila* cells[47] (orange data points Fig. 2e). Although the experimentally known $R_g$ values are for a different cell type, they do indeed fall in the range that we are predicting, suggesting that $R_g$ values from our simulation are reasonable. In Supplementary Fig. 5d, e, we also show the distribution of $R_g$ values.

To independently check the order of magnitude of $R_g$ values, we employed another realistic, detailed model of short chromatin with explicit nucleosomes having entry/exit angles and linker DNA explicitly (Supplementary Fig. 2b and Supplementary Note 1B). We find that

the size of a 1 kb chromatin made of 5 nucleosomes in this model is comparable to what we predict using our basic fine-grained model (Supplementary Fig. 2c), suggesting that the fine-grained model we use is reasonable.

## Coarse-grained chromatin beads are not hard spheres; they overlap impacting bond length and stiffness

Another important quantity is the distance between the centers of two neighboring coarse-grained beads, defined as the bond length $l_{cg}$, depicted in Fig. 2a. In Fig. 3, we present the bond length, its statistics, and the stretching elastic constants derived from it. First, mean $l_{cg}$ values vary depending on the coarse-graining size and genomic location for all the genomic regions we studied. (Fig. 3a, b and Supplementary Fig. 7a, b). Similar to $R_g$, the bond length is also high at certain genomic locations like boundaries of TAD-like domains and low in the domain interiors (Fig. 2c, d and Supplementary Fig. 6). This is essentially the same behavior found in recent experiments[42], which showed that the inter-TAD distances are larger than the intra-TAD distances, consistent with the spatial variation in coarse-grained bead sizes observed in our simulations. Given that a typical chromatin polymer will contain both euchromatic and heterochromatic regions of different compaction, it is instructive to compare extreme sizes of coarse-grained beads to understand the variability one can expect along the genome. For a 5 kb segment ($n_b = 25$), the mean $l_{cg}$ values at different genomic locations vary in the range $\approx 95–120$ nm (heterochromatin) to $\approx 105–150$ nm (euchromatin)−compare $n_b = 25$ in Fig. 3a, b. Interestingly, these values are comparable to the $l_{cg}$ measurements from microscopy experiments that "paint" 5 kb chromatin segments[47]. This also implies that real chromatin will have highly heterogeneous bead dimensions, unlike the prevalent uniform bead size picture. The mean values of bond length, averaged along the polymer contour, are shown as a function of $n_b$ (coarse-graining scale, genome size) for ten different chromatin loci (Fig. 3c). Equivalent coarse-grained bond lengths for SAW and globule are shown as control. Similar to $R_g$, there is some amount of variability in mean $l_{cg}$ across different gene regions; however, they fall within the range of 58–67 nm for 1 kb and 114–132 nm for 10 kb.

How is $l_{cg}$ related to $R_g$? Naively one would expect that $l_{cg} \approx 2R_g$. However, this is not the case; chromatin has $l_{cg} < 2R_g$ because the two nearby polymer segments can "mix"−CG beads can overlap−and have their center of mass locations nearby (Fig. 3d). To quantify the overlap or mixing between the two adjacent coarse-grained beads, we define an overlap parameter $\mathcal{O} = (2\langle R_g \rangle / \langle l_{cg} \rangle)$ (see Fig. 3e and Supplementary Note 1D). If the coarse-grained regions are perfectly spherical and non-overlapping (no mixing) $\mathcal{O} \leq 1$. Imagine two hard spheres of radius $R_g$ connected by a spring. The thermal fluctuations will result in the average inter-bead distance (the equivalent of $l_{cg}$ here) being slightly larger than $2R_g$ and $\mathcal{O} < 1$. As a control, one can see that for SAW, $\mathcal{O} < 1$ and is nearly independent of coarse-graining scale ($n_b$). For euchromatin and heterochromatin $\mathcal{O} > 1$ (i.e., $l_{cg} < 2R_g$), implying the mixing of adjacent polymer segments. We also find that $\mathcal{O}$ depends on the coarse-graining scale − overlap is high at larger $n_b$ values. Since $l_{cg} \neq 2R_g$, should one consider $l_{cg}$ as the size (diameter) of an effective coarse-grained bead, or $2R_g$? This is a relevant question for coarse-graining; given the fact that the beads can overlap, we propose that $l_{cg}$ may be considered as the effective diameter of a coarse-grained bead since it is the effective bond length. We also computed $\mathcal{O}$ as a function of genomic location (Supplementary Fig. 8a,b). Comparing the overlap with the contact map shows that the overlap at the boundary of TAD-like domains is smaller than the domain interior (Fig. 2c, d and Supplementary Fig. 6). While the above quantity measures the overlap between neighboring regions ("bonded" CG beads), any two chromatin regions residing far away along the polymer contour ("nonbonded" CG beads) can also overlap. To quantify this overlap, we first computed the probability distribution of 3D distance $r_{ij}$ between any two CG

beads, $P(r_{ij})$ (Supplementary Fig. 8c). The probability that $r_{ij} < l_{cg}$ is a measure of overlap among far away beads (see Supplementary Fig. 8d).

Going beyond the average size, we computed the bond length distribution $P(l_{cg})$ that has all the information about the fluctuation and higher moments. As a control, for the ideal polymer chain, $P(l_{cg})$ from the simulation matches with the analytical relation proposed by Laso et al.[76] (Supplementary Fig. 7c). We then plot $P(l_{cg})$ for different chromatin loci for different coarse-graining sizes (Fig. 3f and Supplementary Fig. 7d, e). From the distribution, we can derive an effective potential energy $V(l_{cg}) = -k_B T \ln P(l_{cg})$ with which two neighboring beads interact. Even though the distribution is not perfectly Gaussian, a measure of the elastic constant of the interaction can be computed from the inverse of the standard deviation. Hence, we define an effective spring constant between two neighboring CG beads as $K_{cg} = \frac{k_B T}{\langle l_{cg}^2 \rangle - \langle l_{cg} \rangle^2}$, where the angular brackets indicate the average computed using $P(l_{cg})$. In Fig. 3g, we plot $K_{cg}$ for different coarse-graining sizes and various gene loci.

The spring constant is scale dependent−it decreases as the coarse-graining size increases. For most gene regions, the spring constant values appear to saturate at a large coarse-graining scale, unlike the SAW polymer. The $K_{cg}$ value for large $n_b$ is in the range $(0.1–1) k_B T/\sigma^2$, which is $\approx (1–10)$ pN/μm. This value is roughly comparable to some of the experimentally measured values from pulling long chromatin under certain in vitro conditions[77]. Note that, in contrast to pulling experiments where external forces can disrupt protein-mediated interactions, our estimate of the spring constant arises purely from thermal fluctuations and is thus expected to be a reliable signature of chromatin flexibility. Moreover, this is for relatively more dynamic MESc; hence the chromatin stretching stiffness obtained here will be less than that from the pulling experiments of the full-length mitotic chromosomes[78].

The spring constant above is presented in units of $k_B T/\sigma^2$ where $\sigma$ is the size of a 200 bp NL bead. However, in coarse-grained polymer simulations, one uses $K_{cg}$ in units of $k_B T/l_{cg}^2$. Since $l_{cg}$ also depends on CG size ($n_b$), the spring constant has a non-trivial behavior and is presented in Fig. 3h. This gives a very useful range of numbers that can be used in all future coarse-grained simulations as $K_{cg} = 5–10 k_B T/l_{cg}^2$ for coarse-grained beads of size 1–20 kb.

## Predicting angle distribution and stiffness of coarse-grained chromatin segments

How flexible is a chromatin polymer segment? Do chromatin polymer segments have intrinsic curvature? While we understand the bend-ability of DNA reasonably well, we know very little about the bending elastic behavior of chromatin. From the large ensemble of structures that we have produced, consistent with nucleosome level Micro-C data, we computed the distribution of the bond angle ($\theta_{cg}$)−angle between two neighboring bonds connecting three consecutive CG beads (see Fig. 4a top), and the dihedral angle ($\phi_{cg}$)−angle between two neighboring planes formed by three consecutive bond vectors (Fig. 4a bottom).

The bond angle is defined as $\theta_{cg} = \pi - \alpha$, where $\alpha = \cos^{-1}(\hat{l}_{cg}^i \cdot \hat{l}_{cg}^{i+1})$, and it can take any value in the range $[0, \pi]$ (see Supplementary Note 1D). As a control, we computed the angle and its distribution for an ideal chain, and our results match well with the known analytical answer[76] for $P(\theta_{cg})$ for different coarse-graining sizes (Supplementary Fig. 9a). Then we computed the distribution of angles $P(\theta_{cg})$ for chromatin segments in different epigenetic states. As shown in Fig. 4b, for an ideal chain, even with no coarse-graining ($n_b = 1$), the distribution has a shape given by $P(\theta_{cg}) = \frac{1}{2}\sin(\theta_{cg})$[76,79]. This is due to the geometric measure, and it implies that when $\theta_{cg}$ is near 90°, a large number of configurations are possible (having different azimuthal angles), while there is only one possible configuration for extreme cases if $\theta_{cg} = 0°$ and

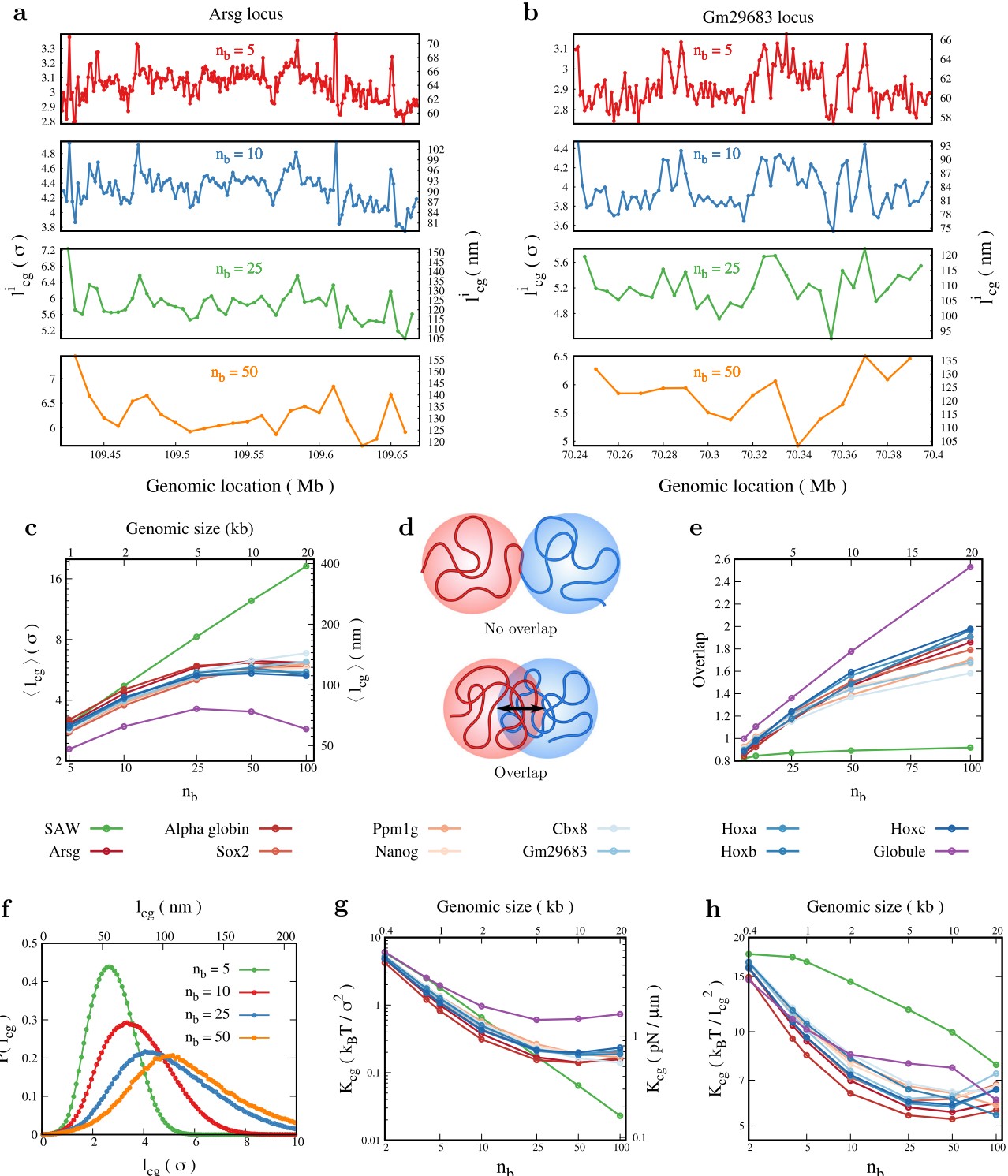

**Fig. 3 | Chromatin as bead-spring chain: bond length, spring constant and overlap. a, b** Spatial variation of average bond length ($l_{cg}^i$) for (**a**) euchromatic Arsg locus and (**b**) heterochromatic Gm29683 locus is plotted for different coarse-graining sizes ($n_b$). **c** Average $l_{cg}$ for different coarse-graining sizes showing the range of values possible from SAW to globule; presented in two different units in $Y_1$ and $Y_2$ axes. **d** Schematic showing two nearby long polymer segments "mixing" (or not mixing) in 3D space resulting in overlap (or no overlap) of coarse-grained (colored) beads. **e** A parameter that quantifies the extent of overlap is plotted for different $n_b$ values. **f** Distribution of $l_{cg}$ for euchromatin region. **g, h** The effective spring constant ($K_{cg}$) quantifies the fluctuation between neighboring coarse-grained beads for chromatin segments in different epigenetic states, presented in two different units. CG simulations would need units in (**h**). Source data are provided as a Source Data file.

$\theta_{cg} = 180°$. Hence, to have a better understanding of the system, we also plot the corresponding probability density defined as $\tilde{P}(\theta_{cg}) = P(\theta_{cg})/\sin(\theta_{cg})$ in Fig. 4c. For the ideal chain, with $n_b = 1$, $P(\theta_{cg})/\sin(\theta_{cg})$ is a flat curve (uniform distribution) reiterating the fact that the ideal chain is unbiased, and all configurations are equally likely. For SAW, the excluded volume would ensure that configurations with $\theta_{cg} \approx 0$ are not possible, and there is a natural bias towards extended configurations ($\theta_{cg} > 90°$).

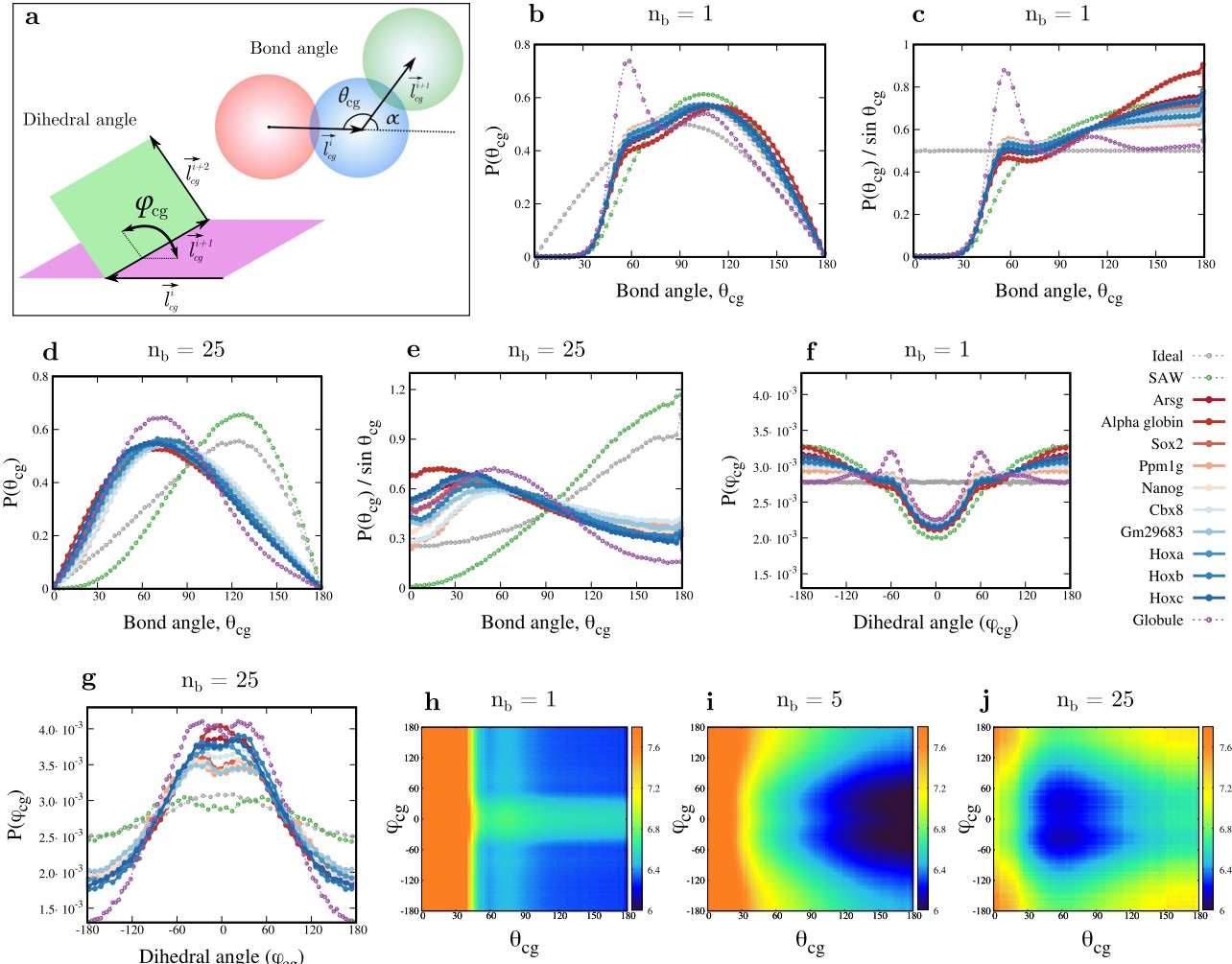

**Fig. 4 | Angle distributions of chromatin segments revealing bendability.**
**a** Schematic showing the bond angle $\theta_{cg}$ and dihedral angle $\phi_{cg}$ between coarse-grained beads. **b**, **c** Distribution of angles for different chromatin loci, for the fine-grained model ($n_b = 1$) with nucleosome-linker (200 bp) resolution (**b**) $P(\theta_{cg})$, and (**c**) the distribution with a different measure $P(\theta_{cg})/\sin(\theta_{cg})$ are shown. **d**, **e** Similar angle distribution for the coarse-grained chromatin model with one bead representing 5 kb ($n_b = 25$) of chromatin is shown in (**d**) $P(\theta_{cg})$ and (**e**) $P(\theta_{cg})/\sin(\theta_{cg})$.

**f**, **g** The distribution of dihedral angles for different chromatin loci for (**f**) the fine-grained model ($n_b = 1$) and (**g**) coarse-grained chromatin with one bead representing 5 kb ($n_b = 25$) of chromatin. **h**–**j** ($\theta_{cg}$- $\phi_{cg}$) energy plot for the Ppm1g locus for (**h**) the fine-grained model ($n_b = 1$), (**i**) CG chromatin with one bead representing 1 kb chromatin ($n_b = 5$) and (**j**) CG chromatin with one bead representing 5 kb chromatin ($n_b = 25$). Source data are provided as a Source Data file.

The emergence of preferred inter-nucleosome angle from folded chromatin configurations: Our results here describe angle distribution for different of chromatin loci. For all the chromatin loci we simulated, at the nucleosomal (fine-grained, $n_b = 1$) resolution, a new peak emerges near $\theta_{cg} \approx 60°$ (Fig. 4b). The deviation from the ideal chain and SAW emerges due to intra-chromatin interactions. Since we do not impose any preferred angle in the fine-grained model, this population with angles near 60° emerges purely from the packaging, based on the contact probability map. To understand this better, we deconvoluted the $P(\theta_{cg})$ distribution and represented it as a sum of two Gaussian distributions giving us two populations having mean values near ≈60° and ≈110° (Supplementary Fig. 9b). Comparing the widths of the two populations suggests that the distribution with mean ≈60° is 2–3 times stiffer than the population with mean ≈110°. For the highly folded globule, the peak at ≈60° is even more prominent, suggesting that tighter packaging could result in a population with ≈60° angles. In contrast to the prevalent notion of smaller angles around ≈60°, our analysis shows that, at least in the case of mESC chromatin, there is a prominent signature of two sub-populations of angles, one highly folded and one extended, for $n_b = 1$ (Fig. 4b, c).

Next, we examined the angle distribution of a coarse-grained chromatin polymer (Fig. 4d, e). A lesser-discussed fact about polymers (even for the ideal chain) is that, when coarse-graining is performed ($n_b > 1$), the angle distribution gains a bias (or a shift), with a preference emerging for the larger $\theta_{cg}$ angles ($\theta_{cg} > 90°$) (See refs. 76,79 and Supplementary Fig. 9a). The SAW polymer has an extra bias towards extended angles as smaller angles are disfavored due to excluded volume effects.

For coarse-gained chromatin, the angle distributions deviate a lot from the ideal chain and SAW, displaying a preferred intrinsic angle around $\theta_{cg}^0 \approx 60°$ for a coarse-graining scale of 5 kb (Fig. 4d, e). For chromatin loci, consistent with ideal chain and SAW, coarse-graining initially shifts the angle distribution towards larger angles (see $n_b = 5$ in Supplementary Fig. 9c–f). However, for larger coarse-graining, long-range intra-chromatin interactions, such as TAD-forming loops, fold chromatin and shift the distribution towards smaller angles (see $n_b = 10, 50$ in Supplementary Fig. 9c–f).

The effect of coarse-graining and deviation from ideal/SAW chain behavior is visible in $\tilde{P}(\theta_{cg})$ distribution as well (Fig. 4e). While the direct experimental readout of angles (e.g., via imaging) would yield $P(\theta_{cg})$, the scaled $\tilde{P}(\theta_{cg})$ is what would be useful for simulations; one can define an

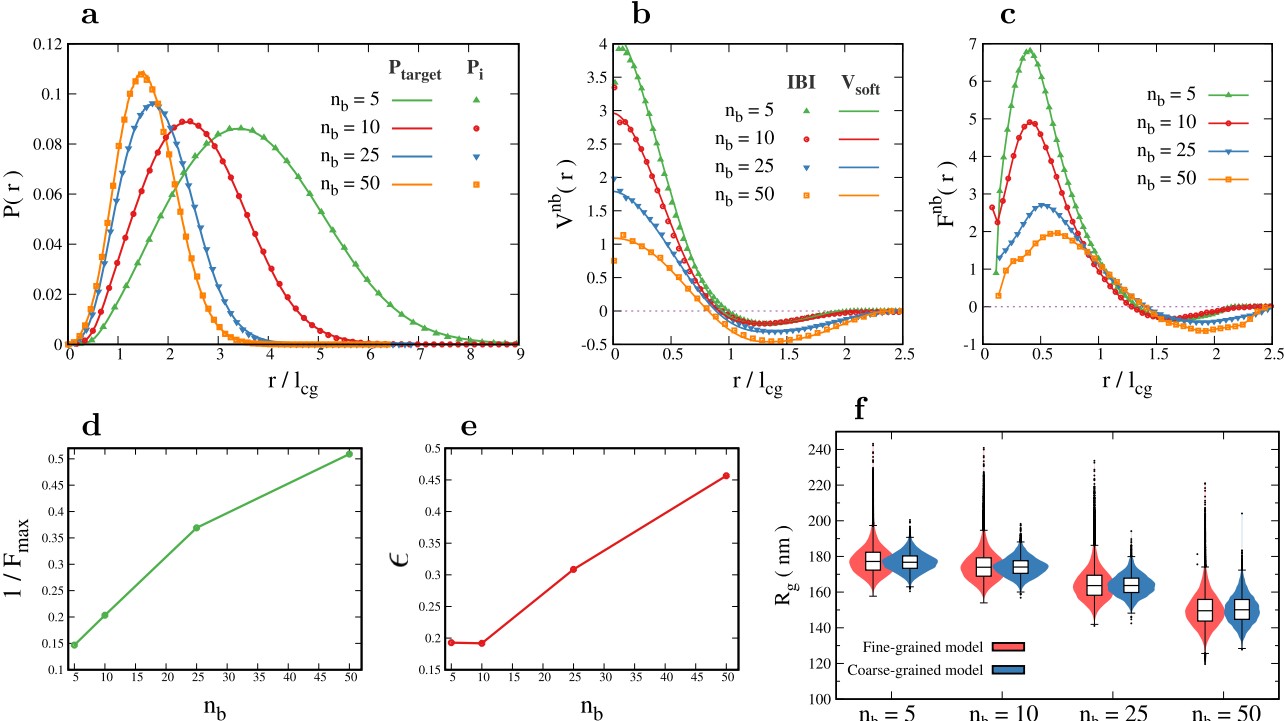

**Fig. 5 | Soft inter-bead potential energy for coarse-grained chromatin beads.**
**a** The distribution of distances between all non-bonded beads obtained from the iterative Boltzmann inversion (IBI) method after convergence $P_i(r)$ (points) matches well with the fine-grained chromatin target distribution $P_{target}(r)$ (solid lines) for appropriate levels of coarse-graining. **b** The non-bonded potential ($V^{nb}$) obtained from the IBI method (points) and the functional form for the soft potential $V_{soft}(r)$ (solid lines using Eq. (2); see parameters in Supplementary Table 2) for different values of $n_b$. **c** The force derived from the potential in (c) for different values of $n_b$. **d** The inverse of the maximum value of the force as a function of $n_b$ as a measure of softness. **e** The depth ($\epsilon$) of the $V^{nb}$ potential is plotted as a function of coarse-

graining size. **f** The radius of gyration measured from the coarse-grained simulation with soft potential is compared with the $R_g$ from the fine-grained model for corresponding levels of coarse-graining. The violin plots show the distribution of data and the box plots on top depict the 25–75th percentiles, with the middle line denoting the median. The whiskers extend to 1.5 times the interquartile range, and outliers are indicated with dots. $n = 60,000$ independent sample polymer configurations were used in fine-grained model while $n = 5000$ independent sample polymer configurations were used in coarse-grained model for all values of $n_b$. Source data are provided as a Source Data file.

effective bond angle potential $V(\theta_{cg}) = -k_B T \log \tilde{P}(\theta_{cg})$[79]. The $V(\theta_{cg})$ curves for chromatin segments have a well-defined minimum at preferred angles, which depends on the coarse-graining scale, $n_b$ (Supplementary Fig. 10). One can also compute the effective bending "stiffness" of chromatin segments by comparing the inverse of the standard deviation around the local maxima of $\tilde{P}(\theta_{cg})$ or by equating $V(\theta_{cg}) = \frac{k_b}{2}(1 + cos(\theta_{cg} - \theta_{cg}^0))$. We find that bending stiffness is of the order of thermal energy $k_b \approx k_B T$ for all coarse-graining scales. This suggests that the chromatin polymer is not highly stiff and explores a wide range of angles. This is consistent with the emerging notion that chromatin is highly dynamic[80–83], and has high cell-to-cell variability.

We examined how the angles vary along the genomic locations. Similar to $R_g$ and $l_{cg}$, angles too have heterogeneity along the genome (Fig. 2c, d and Supplementary Fig. 9g, h). A comparison of the average angle for different genomic locations reveals that there are higher angles near TAD-like domain boundaries and lower angles in the interior of the domains. This spatial variation of chromatin properties could be important for understanding and reconstructing chromatin configurations.

Dihedral angle distribution: The distributions of the dihedral angles $P(\phi_{cg})$ for different chromatin/polymer segments for fine-grained ($n_b = 1$) and coarse-grained ($n_b = 25$) level are shown in Fig. 4f, g. For the fine-grained model ($n_b = 1$), the ideal chain (control) has a uniform angle distribution as expected; the SAW polymer has a dip near $\phi = 0$ indicating self-avoidance/steric hindrance (Fig. 4f). Even for an ideal chain, the coarse-graining leads to non-trivial changes in the $\phi$ distribution where a preference for smaller dihedral angles arises. For the fine-grained chromatin and globule, due to high folding, the

probability of obtaining smaller angles ($\phi$ near zero) increases, and larger angles become rarer compared with the SAW polymer. Folding also leads to a peak near $\phi \approx 60°$, which is prominent for the globule.

A preference for smaller $\phi$ values appears on coarse-graining, similar to the Ideal chain. At the same time, the folding via long-range intra-chromatin interactions results in the formation of peaks near $\phi \approx \pm 60°$. Both of these effects together define the coarse-grained $\phi$ distributions (Fig. 4g). Similar to the bending angle distribution, the dihedral angle distributions are very broad, implying a weak angle stiffness.

Similar to $V(\theta_{cg})$, we define an effective dihedral potential $V(\phi_{cg}) = -k_B T \log P(\phi_{cg})$ (see Supplementary Fig. 11). Assuming that the distributions of $\theta_{cg}$ and $\phi_{cg}$ are independent, we have plotted the heatmap of $V(\theta_{cg}, \phi_{cg}) = V(\theta_{cg}) + V(\phi_{cg})$ in the ($\theta_{cg} - \phi_{cg}$) plane (see Fig. 4h–j). Here the color-bar represents the energy $V(\theta_{cg}, \phi_{cg})$ values in $k_B T$ units. For the fine-grained model, low values of $\theta_{cg}$ and $\phi_{cg}$ are penalized due to self-avoidance (Fig. 4h). This effect reduces with coarse-graining. For lower coarse-graining, higher $\theta_{cg}$ and intermediate $\phi_{cg}$ values are preferred (Fig. 4i), while for higher coarse-graining $\theta_{cg}$ in the range 50°–90° and $\phi_{cg}$ values close to ±60° are favored (Fig. 4j). This again shows that angle preferences for chromatin are scale-dependent—depending on the coarse-graining scale, the preferred values vary considerably.

## Determining optimal soft inter-bead potential and simulating a coarse-grained chromatin

This work systematically estimates the size of coarse-grained beads, their fluctuations, the overlap among the beads due to the mixing of

polymer segments, and the distribution of bond and dihedral angles. Here we integrate these quantities to simulate a coarse-grained chromatin polymer and predict 3D size or $R_g$ measurable in microscopy experiments. While the spring constants and bead sizes ($l_{cg}$) can be directly used from our results discussed so far, we lack the non-bonded interaction potential that would ensure appropriate compaction. Therefore, we perform an iterative Boltzmann inversion (IBI) to determine the form of an inter-bead soft potential that would achieve the 3D distance distribution consistent with our original fine-grained simulation (see Supplementary Note 1C and Supplementary Fig. 12).

We implemented the iterative Boltzmann inversion method for the Arsg locus that we studied using the fine-grained model. Since the overlap (Fig. 3e) depends on the level of coarse-graining, it is expected that the potential energy would also depend on $n_b$. Hence, for each $n_b$, we simulated a coarse-grained bead-spring polymer with $N/n_b$ beads connected by harmonic bonds with equilibrium bond length $l_{cg}$ and spring constant $K_{cg}$ taken from Fig. 3c, h. Starting with a flat inter-bead potential energy $V_{i=0}^{nb} = 0$, at each step of iteration $i$, we simulated the polymer until equilibrium and updated this potential using the relation:

$$V_{i+1}^{nb}(r) = V_i^{nb}(r) + \alpha(r)\, k_B T\, \ln\left(\frac{P_i(r)}{P_{\text{target}}(r)}\right). \tag{1}$$

Here $P_i(r)$ is the steady-state distribution of distances between all pairs of non-bonded beads. The distribution was compared with the known distance distribution, a target distribution $P_{\text{target}}(r)$ for the corresponding level of coarse-graining from our fine-grained model. $\alpha(r) = 0.2\, e^{-r^2/2}$ is taken as a decaying function to ensure that the resulting potential is short-range. We checked for the convergence of the algorithm by computing the Kullback-Leibler Divergence between the target and CG model distance distributions (see Supplementary Note 1C and Supplementary Fig. 13a). In other words, for each $n_b$, we have computed a soft potential energy function between CG bead pairs and used it to perform CG polymer simulations that would reproduce 3D distance distribution exactly as we got from our fine-grained model (see Fig. 5a). The resulting potential energy functions $V^{nb}$ for various $n_b$ values are shown in Fig. 5b. We also fit a functional form to this potential (solid line), such that the softness and depth of the potential can be tuned independently. We use the functional form

$$V_{\text{soft}}(r) = \begin{cases} V_0\left[1 - \left(\frac{r}{r_m}\right)^{\eta_1}\right]^{\eta_2} - \epsilon & r < r_m, \\ \frac{1}{2}\epsilon\left[\cos(\mu r^2 + \nu) - 1\right] & r_m \leqslant r < r_c, \\ 0 & r \geqslant r_c. \end{cases} \tag{2}$$

The first part of the equation ($r < r_m$) gives the repulsive part of the potential[84]. Here, $V_0$ controls the height of the potential at $r = 0$ (see Supplementary Fig. 13b), $r_m$ is the position of minima, and $\epsilon$ denotes the depth of the potential. The parameters $\eta_1$ and $\eta_2$ can be tuned to get the desired softness. The second part of the equation ($r_m \leqslant r < r_c$) represents the attractive part of the potential[24,85,86]. This function has the advantage that it can ensure the continuity and differentiablity at $r = r_m$ and $r = r_c$ by tuning the values of $\mu$ and $\nu$ such that the value of potential is $V_{\text{soft}}(r = r_m) = -\epsilon$ and $V_{\text{soft}}(r = r_c) = 0$ (see Supplementary Note 1C and Supplementary Table 2). The negative slope of the corresponding potential $F^{nb} = \frac{-dV^{nb}}{dr}$ is plotted in Fig. 5c. The inverse of the maximum value of the force ($1/F_{max}$) can be used as a measure of the softness of CG beads (Fig. 5d). The depth of the potential captures the effective attractive interaction between a pair of beads (Fig. 5e). The important points to note are: (i) the potential is derived from fine-grained model that is consistent with the Micro-C experimental data. (ii) The potential is highly soft – softer than the typically used LJ potential (Supplementary Fig. 13c). (iii) The potential energy and the two important physical parameters of the potential – softness and the

attractive interaction strength—are scale-dependent. Different levels of coarse-graining have different softness and interaction strength. This is highly relevant for anyone wanting to simulate chromatin as a coarse-grained bead spring polymer.

Finally, we compare the radius of gyration of the chromatin polymer predicted by our CG simulations with our fine-grained model for various levels of coarse-graining (Fig. 5f). The radius of gyration values match with the fine-grained model. Note that this is equivalent to comparing a coarse-grained model simulation results with microscopy experiments that label DNA/chromatin (e.g., methods that "paint" chromatin[47]) with equivalent resolution. The radius of gyration of both fine-grained and coarse-grained polymers decreases slightly with increasing coarse-graining. This is because the distance of a CG bead from the center of mass of the polymer is smaller than the root mean square distance of the fine-grained beads it replaces (see Supplementary Fig. 13d). This also predicts that the overall $R_g$ value of a long chromatin region (made of many painted small segments) will marginally decrease as one increases the length of the labeled (painted) segment. This decrease is of course less than the size of the painted segment ($l_{cg}$). As mentioned elsewhere in this manuscript, we find that the most probable value that we predict for $l_{cg}$ is comparable with the available data from chromatin microscopy experiments that paint 5 kb segments.

## Discussion

This paper addresses a fundamental question in modeling chromatin: what are the properties and parameters of a coarse-grained chromatin polymer, and how do they vary in a scale-dependent manner as we go from the -10 nm nucleosome scale to hundreds of nanometers gene scale, domain scale or micron-sized chromatin scale? Recent papers have given us a good understanding of the scaling laws, TAD formation, roles of phase separation, loop extrusion, and so on[72,82,87–91]. However, we do not understand the physical dimension of loci that we consider a "bead" in simulations, how stretchable chromatin loci are (spring constant), angle flexibility (bendability), how soft the inter-bead potentials are, and so on. We do not know how chromatin compaction ($R_g$), spring constant, bending angle, overlap, etc., depend on the local contact map (e.g., TAD) structures and epigenetic states.

To fill this gap, we used the recently published Micro-C contact map for mESCs and constructed an ensemble of chromatin configurations at 200 bp resolution. These configurations simultaneously satisfy three constraints: (i) they comply with the Micro-C contact probability, (ii) the mean 3D distance values computed from the configurations are comparable with known experiments, and (iii) the size of the 200 bp fine-grained bead (nucleosome + linker) is in a sensible range. We used this set of configurations and systematically coarse-grained them to predict physical properties and parameters relevant to a chromatin polymer bead-spring chain. We have determined the physical dimensions of chromatin loci (bead sizes of chromatin polymer) for ten different mESC gene regions having different epigenetic state characteristics. We have computed the distributions of the inter-bead distances, predicting how stretchable different chromatin loci are and quantifying their spring constants. We have also predicted the bending and dihedral angle fluctuations revealing how bendable chromatin loci are. Our work not only shows the similarity/variability among different loci but also reveals the effect of chromatin heterogeneity along the polymer contour, finding that TAD interior and TAD boundary have different properties and parameters—different CG bead dimensions, average angle values, overlap, etc. Contrary to the prevalent notion, our results show that CG chromatin beads should be modeled as soft particles that can overlap. We then compute the inter-bead soft potential and propose a functional form to quantify the softness. All our predictions reveal how chromatin properties and parameters change in a scale-dependent manner.

The chromatin polymer parameter values that we have predicted −bead sizes, spring constants, angle distributions, overlap/softness, etc.−are essential for anyone wanting to simulate chromatin polymer. We provide a comprehensive prediction of numerical values of all parameters starting with nucleosome resolution data. Moreover, our finding that chromatin polymer parameters depend on the scale one chooses to study is significant. The polymer parameters relevant for 1 kb chromatin are not the same as that for 10 kb or 100 kb chromatin, which is essential to account for in future simulations. We also argue that many of these parameters (like overlap) are crucial for predicting 3D distance accurately. We have determined an effective inter-bead potential via an iterative Boltzmann inversion method. We used all of these CG results to compute the $R_g$ of a chromatin locus. In other words, our claim is: we have computed the relevant parameters for a polymer simulation of mESC chromatin at different scales. Anyone can use our parameters, simulate coarse-grained chromatin satisfying contact probability, and predict average 3D distances reasonably well within the region-to-region variability we show. Our work has biological significance for connecting chromatin structure to function. Many of the biological processes like recombination, DNA breakage/repair, enhancer-activation, and spreading of histone modifications occur at the scale of nucleosomes. The 3D structure we predict at nucleosome resolution is crucial for understanding these functional aspects. Our work connects the coarse-grained picture (100 nm to μm scale experiments having a few kb or Mb resolution) with a nucleosome-resolution picture and will enable Hi-C or Microscopy experiments to extrapolate and predict nucleosome-level structure. This is highly relevant for understanding the biological functions that occur at nucleosome resolution.

While building the fine-grained model, we made minimal assumptions. The primary assumption we made is that all the chromatin details (e.g., inter-nucleosome interaction potential, histone tails, etc.) result in deciding the contact probability; generating conformations that satisfy the contact map would implicitly account for the role of various local chemical and structural details. Since we use the Micro-C data with 200 bp resolution, our model (model-I) cannot study details below this resolution. We also employed a model with linker DNA (model-II) and showed that our model-I results are sensible. Using model II, we also examined how the variability in nucleosome positioning would affect the overall size ($R_g$) of the folded chromatin. Apart from studying a fixed linker length of 50 bp, we have also performed simulations choosing linker lengths from a Gaussian distribution to incorporate variability. If there are $5 \pm 1$ nucleosomes in 1 kb chromatin, the mean $R_g$ is roughly the same order of magnitude as we reported (Supplementary Fig. 2d). We have also reported these quantities for different mean linker length values. The difference in mean may represent different chromatin states.

For the ten gene loci we studied, heterochromatic and euchromatic regions have $R_g$, angles, and other quantities in a comparable range. This could be because (i) our study is for an embryonic stem cell where the chromatin could be more open. (ii) The underlying Micro-C data itself shows that heterochromatic and euchromatic regions have comparable contact probability as a function of genomic distance $P(s)$ (Supplementary Fig. 4e). This can also be consistent with the irregular nature of chromatin organization as indicated by the power law decay of $P(s)$. Recent experiments have also indicated that heterochromatin can be diverse, and euchromatin can get highly folded due to multiple loops, resulting in similar compaction and other physical properties[48,92,93].

Our work predicts the mean values and variability in bead sizes and other physical properties like elasticity and bendability. It has been suggested that the variability in thickness and flexibilities could affect the chromatin properties below 100 kb[94]. This implies that the variability we find may be relevant since many of the enhancers and promoters can be within 100 kb[40,95]. However, note that, apart from

variability, we predict the average bead size, $l_{cg}$, $K_{cg}$, etc.; the change in the average value would affect measurable quantities at all length scales.

One of the important results of our work is the inter-bead soft potential and the quantification of overlap. Very high-resolution models or models that used sub-beads to represent a larger CG bead would have some signatures of overlap[21,72]. However, most of the current coarse-grained simulation studies use the Lennard-Jones potential for inter-bead interactions, and it quickly goes to infinity with negligible softness. Unlike earlier models[12,13,21,72,84], here we derive the functional form of the soft-potential starting with nucleosome-resolution contact map data and quantify the softness in a scale-dependent manner. One of the concerns regarding the soft potentials is that it may allow chain crossing leading to incorrect dynamics. However, recent experiments show that chain crossings are indeed present, and topoisomerase activity is required to remove these crossings and have entanglement-free interphase chromosomes[96]. This implies that more accurate dynamics would require the presence of enzymes like topoisomerase that actively regulate chromosome topology in terms of entanglements. This may be an essential feature necessary to study dynamics in coarse-grained models.

Experimental tests of our predictions: We simulate the fine-grained model (scale -10−20 nm) and predict quantities at a much larger scale (-100 nm−μm) that can be measured in experiments. Our predictions of the radius of gyration, bond length, and 3D distances can be tested using microscopy experiments, and we have compared some of them in Fig. 1e and Fig. 2e. Combining biochemistry and microscopy, recent studies have proposed methods to "paint" chromatin segments (size ≈5 kb or higher) and trace the chromatin contour. This method allows one to test many polymer predictions, including coarse-grained inter-bead distances and angle fluctuations. Even though the experimental data is not available for the mESC segments that we simulated, we compared our predictions with the available data, and we found that the most probable value that we predict for $l_{cg}$ is comparable with the measured data[47]. We also find that our prediction of the fluctuation of the angles−width of the angle distribution−is comparable to the experimentally measured values. Such experiments may be performed for the mESC gene regions we simulated to compare with our predictions. Future experiments could also test how these values change as one changes the segment size indicating how bead sizes and bendability would vary with the choice of the coarse-graining scale. Future experiments may also measure spring constants at different scales, either through measuring chromatin segment fluctuations or doing pulling experiments at various scales. All of our predictions can be tested using microscopy, chromatin pulling, and other biophysical experiments.

It must be stated that the whole of our analysis is based on the Micro-C data for the mESCs from Hsieh et al.[35]. Hence the numbers emerging from this study would represent embryonic stem cell chromatin. In the future, analysis can be further extended to study various other cell types as new data emerge. The future direction is also to understand the role of nucleosome positioning heterogeneity and assembly/disassembly/sliding kinetics. It requires a much more detailed polymer model[97] and a model to understand how chromatin conformation capture contact maps are influenced by the heterogeneity of nucleosome organization.

## Methods

Model-I. Fine-grained chromatin model with 200 bp resolution chromatin: Our basic model is the fine-grained chromatin polymer model with 200 bp resolution, constructed based on the publicly available Micro-C data for mouse embryonic stem cells (mESCs)[35,36]. The polymer is made of $N$ spherical beads, having the size of 200 bp chromatin (diameter $\sigma$), with nearest-neighbor connectivity via harmonic springs and self-avoiding interaction via the repulsive part of the

Lennard-Jones potential (see Fig. 1a, Supplementary Note 1A). Since each bead consists of a nucleosome and 50bp linker DNA, we call the bead a "nucleosome-linker" (NL) bead. To generate an ensemble of configurations consistent with Micro-C data, we connected (brought into proximity) bead pairs $i$ and $j$ with the experimentally observed contact probability $P_{ij}$ in a two-step process. First, we defined a set of prominent (strong) contacts of the Micro-C contact map (see Supplementary Note 1A)[71]. Taking only the prominent contact probability values, we inserted harmonic springs between bead pairs $i$ and $j$ if $r_n < P_{ij}$, where $r_n$ is a uniformly distributed random number between 0 and 1. Using this procedure, we generated 1000 independent polymer configurations and equilibrated them using Langevin simulations with LAMMPS[98]. We defined "prominent contacts" as follows[71]: Since the contact map depends only on $|i - j|$ for homogeneous polymers, we took the set of all $P_{ij}$ values for a given $|i-j|$ and computed their mean and standard deviation. If $P_{ij}$ was at least one standard deviation larger than the mean, we considered it as a prominent contact (see Supplementary Note 1A). Prominent contacts are defined for each $|i - j|$ line in the matrix (line parallel to the diagonal representing all equidistant bead pairs). Bonding prominent contacts ensured that all actively acquired far-away contacts (e.g., contacts via loop extrusion) were present.

In the second step, going beyond the prominent contacts, our aim is to insert contacts in the $P_{ij}$ fraction of the configurations (out of the 1000 configurations) for each bead pair $(i, j)$. To achieve this, we started with the ensemble of equilibrated configurations from step-1 and inserted harmonic springs between beads $i$ and $j$ in the $P_{ij}$ fraction of configurations whose 3D distances ($r_{ij}$) are the smallest (see SI). This system was then equilibrated using Langevin simulations with LAMMPS. While the first step ensured that the strong contacts formed via events like loop extrusion were established, the second step ensured that all bonds closer in space would have priority in forming protein-mediated contacts.

We have used the minimal fine-grained model that accounts for polymer connectivity, self-avoidance, and contact probability. The assumption here is that all other properties of the fine-grained polymer (like inter-nucleosome interactions and stiffness) lead to the experimentally observed contact probability, which we have ensured. Our model generates all possible polymer configurations such that the experimentally known constraint of the contact map is satisfied.

Size of a 200 bp chromatin bead ($\sigma$): Since a 200 bp chromatin bead is bigger than a nucleosome, its size has to be greater than the size of the nucleosome (11 nm)[3]. As geometrically shown in Supplementary Fig. 2a, since two neighboring nucleosomes are connected via a rigid 50 bp linker DNA, the distance between them can be $\approx$28 nm. However, two far-away nucleosomes can come as close as 11–12 nm (with histone tails and other bound proteins). Hence, on average, one expects an effective size $\approx$20 nm. In the Results section, we have shown that when $\sigma = 21$ nm, the 3D distances and $R_g$ values match well with experimental data. This is sensible considering the linker length and that a typical nucleosome in vivo will likely be covered by several enzymes/proteins like acetyl/methyl transferases, HMG, HP1, remodelers, etc. This is also consistent with the earlier observation that $\sigma = 25$ nm for 250 bp beads[72].

Since Model-I did not have explicit linker DNA, we also simulated short chromatin with nucleosomes, explicit linker DNA, and entry-exit angles between nucleosomal DNA. In this detailed Model-II, the chromatin polymer has two types of beads—linker DNA bead and nucleosome bead (see Supplementary Note 1B)[99]. In the Results section, we have compared the radius of gyration of chromatin segments from Model-II and the first fine-grained model. This also suggests that our $\sigma = 21$ nm value is indeed reasonable.

## Reporting summary

Further information on research design is available in the Nature Portfolio Reporting Summary linked to this article.

## Data availability

Published Micro-C data[35] used in this study is available at the Gene Expression Omnibus (GEO) database with accession number GSE130275. Relevant data generated from this study are included in this article's Figures, text, and supplementary information. Source data are provided with this paper.

## Code availability

All the simulations, analysis and visualization in this study were performed using publicly available software packages or custom codes. LAMMPS (16 March 2018) version was used to perform Langevin Dynamics simulations. VMD version 1.9.3 was used for visualization of 3D polymer configurations and computation of dihedral angles. Custom codes were used for all other analysis. All the codes required to perform the simulations are available in the repository: https://github.com/sangramkadam/chromatin_coarse_graining[100].

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

## Acknowledgements

R.P. acknowledges useful discussions with Xavier Darzacq, Leonid Mirny, Daniel Jost, Geeta Narlikar, and Marc Marti-Renom. We acknowledge useful discussions with Vladimir Teif, Mayuri Rege, PB Sunil Kumar, Madan Rao, and Gaurav Bajpai. S.K. acknowledges fellowship support from the CSIR, India, and KK acknowledges iPDF support from IIT Bombay. We acknowledge funding from the Department of Biotechnology, India (Grant number: BT/HRD/NBA/39/12/2018-19). We also acknowledge the National Supercomputing Mission (NSM) for providing computing resources of 'PARAM Brahma' at IISER Pune, which is implemented by C-DAC and supported by the Ministry of Electronics and Information Technology (MeitY) and Department of Science and Technology (DST), Government of India. R.P. acknowledges support from Sunita Sanghi Centre of Aging and Neurodegenerative Diseases, IIT Bombay.

## Author contributions

R.P., S.K., and K.K. conceived the project. S.K., R.P., and K.K. designed the project with inputs from M.K.M. S.K. performed the research with guidance from R.P. and M.K.M. S.K. developed the codes, performed simulations and analyzed the data with input from all authors. V.M. developed codes and simulated the Model-II with inputs from S.K., S.D. and R.P. All authors contributed ideas and participated in the scientific discussion. S.K. and R.P. wrote the initial draft of the paper with inputs

from M.K.M. All authors edited and prepared the final version of the paper.

## Competing interests

The authors declare no competing interests.
