## [Peer Review File · Nature Communications]

Predicting scale-dependent chromatin polymer properties from systematic coarse-grainingReviewers' comments:

Reviewer #1 (Remarks to the Author):

In this manuscript, the authors developed a "fine-grained" polymer model for chromosomes in which the contact information from Micro-C data is utilized to construct the energy function. Using the model, the author investigated the scale-dependent polymer properties of the chromosome, including bead dimension and elastic parameters. The authors conclude that it is necessary to incorporate soft inter-bead interaction when developing coarse-graining models for chromosomes. The idea that coarse-grain beads for chromatin can overlap is reasonable and make a lot of sense. I would say that it is not fully new as it is well known in the polymer literature that the effective potential (potential of mean force) between the center of mass of two polymer chains is non-divergent at zero distance. Still, a systematic analysis of polymer properties of chromatin such as what is attempted in the present study is valuable in the field. Hence I think this work is in principle suitable for Nature Communications. I do think the manuscript needs some additional discussion and analysis. Please see the comments and questions below.

1. The basic unit of the model is one nucleosome + one linker DNA. This assumes a uniform distribution of nucleosomes along the chromatin fiber. Is there a rationale for this assumption? At the level of 200 bps resolution, it may be important to consider the variability in nucleosome spacing. In particular, euchromatin and heterochromatin may have different nucleosome positioning. And this variability can lead to different polymer properties between euchromatin and heterochromatin, such as bead size and persistence length.
2. The DNA persistence length is 150 bps and the linker DNA is only 50 bps. Then what is the rationale for using a flexible chain model (Eq. 3 in the SI)? If appropriate angle potential is incorporated, I suspect many main results presented will be affected.
3. The overlapping is established by analyzing the distance between the centers of two neighboring coarse-grained beads (Fig.3). In the section "simulating a coarse-grained chromatin", the authors use a pairwise soft LJ potential to model the softness. However, this potential is for non-bonded coarse-grained beads. To be consistent, the "softness" should also be investigated for pairwise inter-beads interaction.
4. One of the interesting questions in modeling chromatin is what are the polymer characteristics of heterochromatin and euchromatin? The authors show that the effective 5kbps bead size is 70nm for heterochromatin and 120nm for euchromatin (Fig. 3). This conforms with the traditional view that heterochromatin is more condensed. However, in Fig, 4, it seems that there is no substantial difference between heterochromatin and euchromatin in angle distribution. Does this suggest no obvious difference in stiffness and torsion between heterochromatin and euchromatin fiber? If so, then this seems to favor a disordered picture of chromatin. The difference between euchromatin and heterochromatin mainly comes from the density alone but not the presence of a higher-order chromatin fiber structure.
5. In the section "simulating a coarse-grained chromatin", the authors fine-tune the softness parameter λ and interaction strength ϵ by matching the experimental R_g value and simulated angle distribution (Fig. 5(b,c)). Is the experimental value of R_g calculated from the same region as the Micro-C data used in this study?
6. This is related to question #5. Since the authors attempt to find the effective λ and ϵ in the coarse-grained polymer model by "matching" the fine-grained system in terms of R_g and angle distribution, I believe a more appropriate way is to compare with **simulated** R_g of the fine-grained system instead.
7. Naturally λ and ϵ in the coarse-grained model should also be scale-dependent. For instance, the larger the coarse-grained bead is, the softer it is, and possibly the larger ϵ is. Could the authors provide some comments/discussion?
8. One of the main conclusions of the present study is that it is necessary to incorporate softness in the inter-bead interaction to predict chromatin 3D distances accurately. This is intuitive since the distances between the center-of-mass of two polymer chains can be as close as zero, which leads to a

non-divergent effective interaction between the two chains. However, one must also consider the topological constraints when modeling chromatin. When soft inter-bead interaction is used, the chain in principle can cross which cannot occur because it is a chain. Presumably, Topoisomerase can lead to chain crossing. But whether it is enough for the chain-crossing to occur over the entire chain is debatable. In addition to its effect on the structures, allowing chain crossing may also have a profound impact on the dynamics. Discussions and comments regarding this point should be provided.

minor comments:

1. To what degree are the results shown in Fig. 2,3,4 region-dependent? The results are obtained by using Micro-C map of Chr5: 31.17 Mbps - 31.25 Mbps and Chr 7: 70.24 Mbps - 70.4 Mbps. If other regions are used, how much the results in Fig. 2, 3, 4 will be changed? I suggest that the authors may perform the same analysis for two more regions (one for euchromatin and one for heterochromatin).
2. The curve for the globule model is not shown in Fig.3e
3. In the section "emergence of preferred inter-nucleosome angle from folded chromatin configuration". It states that the chromatin domain investigated have two subpopulations: one highly folded and one extended. The reasoning for this statement is that $P(\theta_{cg})$ can be deconvoluted into two Gaussians (Fig. S5). I have two related questions:
 - a. Fig.4(b) shows that even for the globule model there are two peaks in the distribution and the two peaks are at similar positions as euchromatin and heterochromatin. Does this mean the globule model also has one extended and one folded subpopulations?
 - b. $P(\theta_{cg})$ may be insufficient to show the existence of extended and folded sub-populations. I suggest that the authors calculate more direct measurements, such as the distribution of radius of gyration or end-to-end distance.
4. Please define P_{ij} with an equation in section A in the SI
5. In section A in the SI, is " $P_{ij}^{pr} < r_n$ " a typo? I thought it should be " $P_{ij}^{pr} > r_n$ "
6. In section A in the SI, please describe how the system is relaxed the system at step 2? When the springs contacts are added to the system, the distance r_{ij} can be quite large. In this case, do the authors relax the system using a small timestep?
7. This is related to the previous question. Since WCA potential is used for pairwise interaction, does the steric interaction hinders the "contacts" imposed by E_{ij}^{mc} ? Are all of the contacts realized?

Reviewer #2 (Remarks to the Author):

I have reviewed the manuscript by Kadam, S. and co-authors entitled "Predicting scale dependent chromatin properties from systematic coarse-graining". The manuscript addresses interesting problem from the field of computational biology that is very topical as supported by a lot of recent literature. However, I have serious doubts about the aim of the work and I find it partially confusing. As such I cannot recommend it for publishing in the Nature Comm. Journal. My comments follow.

First of all, I find confusing if the authors do understand the purpose of modelling, where a good model should provide predictions that can be tested experimentally. Here, the authors use experiments in a somewhat opposite way, to do predictions about their model, the intended use of which is not clear, as the model is not further applied to solve any specific experimental problem, so it looks like a half of the story is missing. Also, in the systematic coarse-graining approach, some details of the structure are neglected for the sake of rationalizing the use of computer time and extending the physical time of simulations, perhaps, to a biological region of interest. The choice of the size of DNA portion to be coarse-grained is arbitrary but should be smaller than is the length scale of the studied problem. In Figure 1S, the authors show "Diameter (physical size) of simulation beads estimated for different chromatin segment lengths in various published papers". I really don't understand what the authors mean to say with this illustration, as they show the portions without any context. For example, the manuscript cites works by Mirny et al., who use various values for the size of the bead e.g. 440, 600 and recently 20 bp (Ref. 78), which were of course chosen with a particular purpose.

In the manuscript's abstract the authors argue about the importance of their computational findings, quote "unlike the prevalent notion, our findings argue that coarse-grained chromatin beads must be considered as soft particles that can overlap, and we propose an overlap parameter." But again, what is the model aimed to be used for? So, for example, the work by Liebermann-Aiden, cited in ref. 28, employed the hard-core potential and showed that the scaling of contacts with genomic distance that decay with exponent roughly equal to -1 corresponds to DNA organized as a fractal that is free of knots, that was very important and fascinating up to these days. Now, varying the softness of the beads would yield a different scaling exponent. So, what are the authors going to study with their model? The idea of using the soft beads to model chromatin is not new, for example in the work by Giorgetti et al from 2014, cited as Ref. 10, the authors did a kind of Reverse Monte-Carlo simulations in N^2 parameter space, fitting also the softness of the beads on the experimental contact maps. But, Kadam et al should realize that the work by Giorgetti came out about a year after discovery of TAD's and it was a pioneering explanation of how TAD's are created. Since then, the "triangles" on contact maps had been becoming less fascinating as various works showed that they can be created by multiple ways, like protein linkers (works by Mario Nicodemi), supercoiling (works by Stasiak group), or even just a bond representing cohesin (works by Leonid Mirny, but also for instance j.cell.2015.11.024), but recent simulations by Mirny et al, shed light on how other prominent features observed on Micro-C are created, cited as ref. 78 in the manuscript.

The simulations in the manuscript test various models, in Model-II the authors use an explicit linker with nucleosomes. Conclusions from such the computational experiment are very tricky, as the authors use a simple beaded chain that does not have torsional stiffness, so the DNA at entry and exit points of the nucleosomes is unconstrained, and imposing just an angle would not work. In addition, the simulations of torsionally unconstrained chains possessing larger and smaller beads were already explored by Angelo Rosa (journal.pcbi.1004987) and showed no differences at larger scales.

All in all, I think the work by Kadam, S. et al. objects established coarse-graining approaches and polymer physics scaling laws as they say "prevalent notion", but the authors show no use to it, the employment of soft beads is not new (Georgetti), simulations with explicit linker DNA miss torsional stiffness and are redundant to the existing works (Rosa), a great deal of important publications is missing (Nicodemi, Mirny, Stasiak, Plewczynski, Rosa), the authors seem to miss important points of the modelling approach and the authors just propose scaling principles for a model whose purpose is unclear. Given the number of issues, I cannot recommend the manuscript for publishing in the Nature Comm Journal. If there was anything aside my judgement to encourage the authors, I like the Figure 2c in the main text of the manuscript.

Reviewer #3 (Remarks to the Author):

In this work, Kadam et al. analyze the relevant properties of coarse-grained beads-on-a-spring models of chromatin, i.e., bead sizes, bond lengths and angles between neighboring beads, and corresponding stretching and bending stiffness. Using available nucleosome resolution micro-C contact data in mESC cells, they first derive ensembles of 3D structures reproducing the data by numerical calculations, exploiting the fact that the chromatin properties are mostly known at the nucleosome scale. Next, by systematically coarse graining the obtained 3D models, they infer the scale dependence of such properties. Additionally, they show trends depending on the chromatin regions, i.e., euchromatin or heterochromatin, and on the genomic positions, e.g., at TADs boundaries compared to TAD interiors. Even though most of such trends are expected (e.g., that gyration radii, bond lengths and angles are larger at TAD boundaries) and in some cases previously observed, there is also some (little) interesting insight into chromatin structure. In particular, the bond angle distribution shows the presence of two peaks, one at the expected nucleosome bond angle and another at larger angles

previously unobserved to my knowledge.

Thus, I think the results are significant for an audience of experts working on chromatin polymer models to build increasingly reliable models and, to a lesser extent, for a broader audience since there is limited biological insight into chromatin structure. The methodology is sound and well explained. However, some questions should be addressed to support the manuscript conclusions as detailed below.

Major comments:

- The authors only model and perform analyses on two very small regions of chromatin, an 80 kb euchromatic region (Chr 5: 31.17-31.25 Mb) and a 160 kb heterochromatic region (Chr 7:70.24-70.4 Mb). Thus, my main concern is that all the obtained results, e.g., the precise quantitative differences found between eu- and heterochromatin, dependence of chromatin polymer properties on coarse-graining, etc. might depend on the specific chosen regions. If so, these results would not give helpful insights for future simulations. The authors could repeat the analyses on other genomic regions (at least a few eu- and hetero- chromatic regions) to show that the results are consistent.

- One key finding is that coarse-grained beads (representing a certain genomic window) in polymer models must be considered as soft particles that can overlap with each other. However, some existing models in literature consider a coarse-grained bead of chromatin as a sequence of sub-beads, thus effectively taking into account the possibility of overlap, as, e.g., in Bianco et al. Nat. Gen. 2018, or in Chiariello et al. Cell Rep. 2020, where, e.g., a 4kb window is composed by a sequence of 16 non-overlapping sub-beads. I think the authors could comment that such models still allow for overlaps between coarse-grained chromatin windows and effectively lead to variable properties, e.g. gyration radii and bond lengths along chromatin.

- The similarity between model and micro-C contact data is measured by Pearson correlation, however it is recognized that it is not a good metric since dominated by the decaying trend of contact probability with the genomic distance. The authors could consider different metrics, e.g., the HiCRep stratum adjusted correlation coefficient [Yang et al. Genome Res 2017], or the Pearson correlation as a function of the genomic distance, or others.

- The authors show that R_g , l_{cg} , θ and overlap values have peaks at TAD boundaries for the euchromatin region (Figure 2c). I wonder if some pattern can also be found for the heterochromatic region (e.g. are the peaks of R_g in the same locations as peaks of l_{cg} , and are they located in specific regions as TAD boundaries?), even though here there is less evident TAD like structure.

- The Conclusions could be expanded to better summarize the main findings.

- It would be useful to add a few more details about the performed polymer simulations in the Supplementary Methods, e.g., what are the initial polymer configurations, how long are the simulations, how many timesteps until the equilibrium is reached.

Minor:

- I think the acronym FG for fine-grained is not ideal since it can be confused with the same used for Fractal Globule model of chromatin

- In Figure 2D caption I would refer to the experimental data as "orange squares" instead of "data points" to avoid confusion. Additionally, the authors could clarify the type of data in the maintext (or also in the caption) because the fact that they are relative to *Drosophila* (and to a repressed region?) only appears in the plot.

- I wonder why the plots in Figure 3 (g) and (h) do not start with $n_b = 1$.

- In Supplementary Methods, first section, "we set $C_{ij} = 1$ if $P_{pr ij} < r_n$," $<$ should be $>$.

Response to reviewers

We thank the reviewers for their insightful and constructive comments. This has helped us to improve the manuscript significantly. Based on the referee comments, we have performed several new simulations and analyses and have revised the manuscript to address all the concerns. Below we address the referee comments and the detailed response to each of the reviewers' points is given below.

Two common major comments were: (1) To perform simulations for many more gene regions and (2) to describe the novelty of the work. During the revision, we performed simulations for many more gene regions. In the revised manuscript, we are presenting data for ten gene loci. We have also done additional analysis to derive an inter-bead soft potential via an iterative Boltzmann method at multiple coarse-graining scales.

We would like to stress the novelty of our work here: (i) First of all, we show that chromatin polymer parameters depend on the scale of coarse-graining. The polymer parameters relevant for 1kb chromatin is not the same as that for 10kb or 100kb chromatin. This is a novel point. (ii) We show that some of these parameters (like overlap) are crucial for predicting 3D distance accurately. We have now determined an effective inter-bead potential via an iterative Boltzmann inversion method starting with the fine-grained model that satisfies the contact map (new section in the revised manuscript). We used all of these coarse-grained results to compute the R_g of a chromatin locus. In other words, our claim is: we have computed the relevant parameters for a polymer simulation of mESC chromatin at different scales. Anyone can take our parameters, simulate coarse-grained chromatin satisfying contact probability, and they can predict average 3D distances reasonably well (within the region-to-region variability that we show). (iii) Beyond the point that our results are essential for CG simulations, our work has biological significance for connecting chromatin structure to function. Many of the biological processes like recombination, DNA breakage/repair, enhancer-activation, and spreading of histone modifications occur at the scale of nucleosomes. The 3D structure we predict at nucleosome resolution is crucial for understanding these functional aspects. Our work connects the coarse-grained picture (100 nm to μm scale experiments having a few kb or Mb resolution) with a nucleosome-resolution picture and will enable Hi-C or Microscopy experiments to extrapolate and predict nucleosome-level structure. This is highly relevant for understanding the biological functions that occur at nucleosome resolution.

We start with nucleosome-resolution data and quantify how the bead size, spring constant, overlap (or softness), bending angle distribution, etc., vary depending on your choice of coarse-graining scale

for multiple genes having euchromatic and heterochromatic modifications. This is novel. There is no work that predicts these parameters starting with nucleosome resolution data with systematic coarse-graining.

Response to reviewer 1:

In this manuscript, the authors developed a “fine-grained” polymer model for chromosomes in which the contact information from Micro-C data is utilized to construct the energy function. Using the model, the author investigated the scale-dependent polymer properties of the chromosome, including bead dimension and elastic parameters. The authors conclude that it is necessary to incorporate soft inter-bead interaction when developing coarse-graining models for chromosomes. The idea that coarse-grain beads for chromatin can overlap is reasonable and make a lot of sense. I would say that it is not fully new as it is well known in the polymer literature that the effective potential (potential of mean force) between the center of mass of two polymer chains is non-divergent at zero distance. Still, a systematic analysis of polymer properties of chromatin such as what is attempted in the present study is valuable in the field. Hence I think this work is in principle suitable for Nature Communications. I do think the manuscript needs some additional discussion and analysis. Please see the comments and questions below.

Thank you for finding our study valuable and suitable for publication in Nature Communications after revision. Moreover, we are grateful for several constructive suggestions and comments. Below we address each of the comments in detail.

1. *The basic unit of the model is one nucleosome + one linker DNA. This assumes a uniform distribution of nucleosomes along the chromatin fiber. Is there a rationale for this assumption? At the level of 200 bps resolution, it may be important to consider the variability in nucleosome spacing. In particular, euchromatin and heterochromatin may have different nucleosome positioning. And this variability can lead to different polymer properties between euchromatin and heterochromatin, such as bead size and persistence length.*

Our starting point is the Micro-C data (Hsieh et al., Mol. Cell, 2020 [1]) having 200bp resolution. Hence we took one bead = 200bp, as that was the best option given the resolution of the contact map. As far as we understand, how nucleosome position variability/dynamics would affect the Micro-C contact map is still being determined, and more experimental and theoretical studies are required on this front.

It needs a different model to introduce nucleosome positioning variability/dynamics into this problem — a model with explicit DNA and histone proteins — and it is beyond the scope of the current work. We hope our work and recent works like Lu et al. [2] will fuel further studies that will take us in that direction. Our own model-II given in the manuscript is a first attempt in that direction, where the linker length and its variability are included. However, incorporating micro-C contact probabilities into model II is not easy; hence, we have incorporated chromatin folding via DNA entry/exit angles in model-II. Using model II, in the revised manuscript, we have examined how the variability in nucleosome positioning would affect the overall size (R_g) of the folded chromatin. Instead of a fixed linker length of 50 bp, we have chosen the linker length from a Gaussian distribution to incorporate some amount of variability. The mean R_g of 1kb chromatin with variable linker length is comparable to what we reported for chromatin with fixed linker length (see SI Fig. S2(c) and (d)). We have also reported these quantities for different mean linker length values (see Fig. R1 and SI Fig. S2(d)). The difference in mean may represent different chromatin states (heterochromatin/euchromatin). In the discussions section, we have described these aspects.

FIG. R1: The radius of gyration of the 1kb chromatin segment simulated using Model II. The linker lengths in the simulation are taken from a Gaussian distribution with the specified mean (μ) and a standard deviation of 10bp. Different mean values could mimic different chromatin states (heterochromatin, euchromatin). The standard deviation represents the variability in the number of nucleosomes. The results are presented for two mean nucleosome angles.

2. *The DNA persistence length is 150 bps and the linker DNA is only 50 bps. Then what is the rationale for using a flexible chain model (Eq. 3 in the SI)? If appropriate angle potential is incorporated, I suspect many main results presented will be affected.*

Our main model (model I) does not have linker DNA explicitly. In our model, one bead is 200bp (including linker DNA), and any two 200bp beads interact according to the Micro-C contact probability. The assumption we have is that the rigidity of DNA and other physical properties of the chromatin contribute to the contact probability. In other words, the effect of rigidity, nucleosome angle, etc., is buried into the contact probability. Another reason for choosing this simple model is to have a model with minimal parameters. The only feature we accounted for is the contact probability, and we feel that all other aspects are indirectly buried in the contact probability. Given the contact probability, we performed the simplest procedure to generate all possible configurations that would satisfy the contact probability constraint.

However, we have also presented a model-II in SI, which indeed uses DNA with appropriate persistence length (see Sec. S1.B). In model-II, we have explicit linker DNA with known persistence length. We have presented a set of basic results (R_g for small chromatin) from this model in SI; model II was simulated to ensure that the order of magnitude values of our main results (model I) are sensible even if we take a different model with more details. However, simulating a long genome with model-II is computationally expensive. Hence we simulated a short region and showed that the size scale we obtained from a detailed model (model-II) is comparable to model-I.

We have discussed these points in the revised manuscript Discussion section.

3. *The overlapping is established by analyzing the distance between the centers of two neighboring coarse-grained beads (Fig.3). In the section “simulating a coarse-grained chromatin”, the authors use a pairwise soft LJ potential to model the softness. However, this potential is for non-bonded coarse-grained beads. To be consistent, the “softness” should also be investigated for pairwise inter-beads interaction.*

We thank the reviewer for making this important point. During the revision, we investigated this aspect in detail. The revised manuscript presents the distribution of distances between all pairs of coarse-grained beads (non-bonded beads). The resulting distribution is presented for different coarse-graining levels ($n_b = 5, n_b = 10$, etc.). See Fig. R2 in this document. The X-axis is scaled with l_{cg} ; hence, the shaded region in the distribution represents the probability of having the 3D distance less than l_{cg} . This is a different measure of overlap computed for non-bonded beads. We have plotted the area of the shaded region separately in Fig. R2(b) and also presented in the Revised MS Fig. S8.

FIG. R2: (a) The distribution of the 3D distance between non-bonded ($|i - j| > 1$) coarse-grained beads is plotted for different coarse-graining levels (Arsg locus). The area under the curve till $r_{ij} < l_{cg}$ (shaded region) can be used as a measure for the overlap or softness of the coarse-grained beads, which is plotted in (b), as a function of n_b , for various regions.

To go one step further, we have used an iterative Boltzmann inversion method and computed the effective soft potential between non-bonded beads. This gives us a better measure of softness. We have described this in detail in the revised manuscript (see Fig. R3 in this document and Fig. 5 in the main manuscript).

4. *One of the interesting questions in modeling chromatin is what are the polymer characteristics of heterochromatin and euchromatin? The authors show that the effective 5kbp bead size is 70nm for heterochromatin and 120nm for euchromatin (Fig. 3). This conforms with the traditional view that heterochromatin is more condensed. However, in Fig. 4, it seems that there is no substantial difference between heterochromatin and euchromatin in angle distribution. Does this suggest no obvious difference in stiffness and torsion between heterochromatin and euchromatin fiber? If so, then this seems to favor a disordered picture of chromatin. The difference between euchromatin and heterochromatin mainly comes from the density alone but not the presence of a higher-order chromatin fiber structure.*

In the revised manuscript, we have computed both R_g and angle distributions for several genes (main manuscript Fig. 2 and Fig. 4). We find that, overall, both R_g and angle distributions have some amount of variability, but the mean falls in a narrow range for the genes we simulated. Both euchromatic and heterochromatic regions have high as well as low R_g values. Similarly, the angle distributions are also comparable for euchromatin and heterochromatin. From this, as the referee suggested, we can conclude that there is no obvious difference in stiffness and torsion between euchromatin and heterochromatin. This indeed seems to favor a disordered

picture. Experimental data and our simulations also show that contact probability as a function of genomic distance $P(s)$ decreases as a power law (Fig. R4 and SI Fig. S4). This is another piece of evidence supporting the disordered picture. If the chromatin was highly ordered, the $P(s)$ would be a highly peaked function. In Fig. S4, we plot $P(s)$ from both the experiment and our simulations for a few euchromatic and heterochromatic regions, and they show similar behavior. This supports our above conclusions.

Recent experiments have also indicated that heterochromatin can be diverse, and euchromatin can get highly folded due to multiple loops, resulting in similar compaction and other physical properties [3–5].

5. *In the section “simulating a coarse-grained chromatin”, the authors fine-tune the softness parameter λ and interaction strength ϵ by matching the experimental R_g value and simulated angle distribution (Fig. 5(b,c)). Is the experimental value of R_g calculated from the same region as the Micro-C data used in this study?*

In the previous version of the manuscript, R_g was computed for a generic region — not for a region with specific Micro-C data — because we used a generic soft potential independent of the location. However, during the revision, we have done a new set of calculations to explicitly compute the nature of the nonbonded interactions from our fine-grained model. We used an iterative Boltzmann inversion method and computed the nonbonded interactions for a specific region. The results are shown in the revised manuscript Fig. R3. Now we compare the radius of gyration of CG polymer simulated using this soft nonbonded interaction with the R_g from the fine-grained model with appropriate coarse-graining (see Fig. R3(f) also see main manuscript Fig. 5). This study ensures that the R_g from the fine-grained Micro-C model is comparable with the R_g of the coarse-grained model with soft potential. These points are described in the revised manuscript section titled “Determining optimal soft inter-bead potential and simulating a coarse-grained chromatin.”

6. *This is related to question #5. Since the authors attempt to find the effective λ and ϵ in the coarse-grained polymer model by “matching” the fine-grained system in terms of R_g and angle distribution, I believe a more appropriate way is to compare with ****simulated**** R_g of the fine-grained system instead.*

In the revised manuscript, we have done this. As mentioned above, we now have a systematic method to obtain appropriate soft potential consistent with the fine-grained model. Using this new soft potential obtained from the iterative Boltzmann inversion method, we have computed the R_g of the coarse-grained polymer and compared it with that of the fine-grained polymer. This comparison is described in the revised manuscript in a new section titled “Determining

optimal soft inter-bead potential and simulating a coarse-grained chromatin.”

FIG. R3: (a) Effective potential energy ($V^{nb}(r)$) between any two non-bonded coarse-grained beads as a function of their separation for different coarse-graining sizes n_b computed using an iterative Boltzmann inversion method (details are described in the revised main manuscript). (b) The force between non-bonded beads $F^{nb}(r) = -\nabla V^{nb}(r)$ (negative of the slope) is plotted for different levels of coarse-graining, (c) The inverse of the maximum value of the force F_{max} plotted as the function of coarse-graining (The maximum force is observed for $r \approx 0.5$). The less the force, the more the softness; this indicates how softness varies with coarse-graining size. (d) The depth of the potential (ϵ) is plotted as a function of coarse-graining. (e) The height of the potential at $r = 0$ (V_0) is plotted as the function of coarse-graining. (f) The radius of gyration of CG chromatin polymer simulated with soft potential is compared with the corresponding polymer obtained by the coarse-graining of the fine-grained model. Note that the overall size of the polymer reduces with coarse-graining as n_b number of beads are replaced by their center of mass positions.

7. Naturally, λ and ϵ in the coarse-grained model should also be scale-dependent. For instance, the larger the coarse-grained bead is, the softer it is, and possibly the larger epsilon is. Could the authors provide some comments/discussion?

Thank you for this question. During the revision process, we computed overlap param-

ter/softness between two non-bonded coarse-grained beads consistent with the fine-grained model (as opposed to a generic soft potential we used in the previous version of the manuscript). We have also extracted an effective soft potential between any two CG beads for different levels of coarse-graining. From this potential, we have extracted a parameter $1/F_{max}$ representing the effective softness (the equivalent of λ). This is indeed increasing with n_b (see Fig. R3(c)). The depth of the interaction potential (ϵ) as a function of n_b is also shown in Fig. R3(d). We have described this in the revised manuscript (see Fig. 5).

8. *One of the main conclusions of the present study is that it is necessary to incorporate softness in the inter-bead interaction to predict chromatin 3D distances accurately. This is intuitive since the distances between the center-of-mass of two polymer chains can be as close as zero, which leads to a non-divergent effective interaction between the two chains. However, one must also consider the topological constraints when modeling chromatin. When soft inter-bead interaction is used, the chain in principle can cross which cannot occur because it is a chain. Presumably, Topoisomerase can lead to chain crossing. But whether it is enough for the chain-crossing to occur over the entire chain is debatable. In addition to its effect on the structures, allowing chain crossing may also have a profound impact on the dynamics. Discussions and comments regarding this point should be provided.*

This is an interesting point. As mentioned above, we have computed an effective soft potential between any two beads based on our fine-grained model in the revised manuscript. This effective potential has high softness, but nonetheless, energy increases as inter-bead distance goes to zero. This will make crossing more difficult. However, how to make CG polymers with soft potentials that cannot cross each other is an interesting question beyond chromatin. One way could be to add an infinite hard barrier at $r \approx 0$ and prevent crossing.

In the specific context of chromatin, recent experiments show that chain crossings are indeed present, and Topoisomerases activity is required to remove these crossings and have entanglement-free interphase chromosomes [6]. This implies that more accurate dynamics would require the presence of enzymes like topoisomerase that actively regulate chromosome topology in terms of entanglements. This may be an essential feature necessary to study dynamics in coarse-grained models.

Minor comments:

1. *To what degree are the results shown in Fig. 2,3,4 region-dependent? The results are obtained by using Micro-C map of Chr5: 31.17 Mbps - 31.25 Mbps and Chr 7: 70.24 Mbps - 70.4 Mbps. If other regions are used, how much the results in Fig. 2,*

3, 4 will be changed? I suggest that the authors may perform the same analysis for two more regions (one for euchromatin and one for heterochromatin).

Thank you for this question. In the revised manuscript, we are presenting new simulation results for several other chromatin regions. Interestingly R_g , spring constant, angle distributions, and all other relevant parameters have some amount of variability. However, the mean (average for a given n_b) falls in a narrow range for all the loci we simulated. We have discussed this in the revised manuscript.

2. *The curve for the globule model is not shown in Fig.3e*

We have added the curve for globule in the revised manuscript.

3. *In the section “emergence of preferred inter-nucleosome angle from folded chromatin configuration”. It states that the chromatin domain investigated have two subpopulations: one highly folded and one extended. The reasoning for this statement is that $P(\theta_{cg})$ can be deconvoluted into two Gaussians (Fig. S5). I have two related questions:*

a. Fig.4(b) shows that even for the globule model there are two peaks in the distribution and the two peaks are at similar positions as euchromatin and heterochromatin. Does this mean the globule model also has one extended and one folded subpopulations?

b. $P(\theta_{cg})$ may be insufficient to show the existence of extended and folded subpopulations. I suggest that the authors calculate more direct measurements, such as the distribution of radius of gyration or end-to-end distance.

Here by “sub-populations,” we mean the sub-populations of individual nucleosomes. That is, some nucleosomes have 60-degree angles, and some have larger angles. One nucleosome can have a 60-degree angle, while the neighboring nucleosome might have a larger angle. Hence, if we compute any macro-state variable (like end-to-end distance or R_g), they may not show two populations because they will average out when we sum.

The fact that the highly folded globule also has two peaks suggests that the 60-degree smaller peak arises from the dense packing – close packing of self-avoiding spheres can lead to 60-degree angles. At the same time, thermal fluctuations can lead to a fraction of extended configurations.

4. *Please define P_{ij} with an equation in section A in the SI*

In the revised manuscript, we have defined the P_{ij} using an equation.

5. *In section A in the SI, is " $P_{ij}^{pr} < r_n$ " a typo? I thought it should be " $P_{ij}^{pr} > r_n$ "*

We have corrected this typo in the revised manuscript.

6. *In section A in the SI, please describe how the system is relaxed the system at step 2? When the springs contacts are added to the system, the distance r_{ij} can be quite large. In this case, do the authors relax the system using a small timestep?*

We begin by initializing the polymer configuration with a self-avoiding walk. We take a step-wise approach to ensure the polymer configuration is stable and does not experience large forces from extra contacts through Micro-C interactions between far-away beads. We introduce extra contacts gradually. First, we start with local contacts between bead pairs i and j that are close together (with a separation of $|i - j| < 5$) and equilibrate the polymer for 10^5 steps. In subsequent steps, we introduce remaining contacts with larger separations ($|i - j| < 10$, $|i - j| < 25$, and so on), relaxing the polymer for 10^5 steps after each step. Once all contacts have been added, we equilibrate the polymer for 6×10^6 time steps. During the last 3×10^6 steps, we sample the data every 50000 steps to analyze the behavior of the polymer in more detail. We have added these details in SI Sec. S1.A.

7. *This is related to the previous question. Since WCA potential is used for pairwise interaction, does the steric interaction hinders the "contacts" imposed by E_{ij}^{mc} ? Are all of the contacts realized?*

Statistically, the steric hindrance can prevent contact in some of the configurations; however, to compensate, in some other configurations, beads do come in contact even if there is no explicit spring between them. These are induced contacts due to nearby springs. Hence, we ensured that the system simultaneously satisfies the three experimentally known quantities (constraints) – (i) The contact probability is satisfied. (ii) The 3D distance is satisfied, and (iii) the size (sigma) of the 200 bp bead (nucleosome + linker) is in a sensible range. Regarding the contact probability, we have computed the stratum-adjusted correlation coefficient to compare the simulation contact map with the experimental map and also compared the contact probability as a function of genomic distance (Table S1 and Fig. S4).

Response to reviewer 2:

I have reviewed the manuscript by Kadam, S. and co-authors entitled "Predicting scale dependent chromatin properties from systematic coarse-graining". The manuscript ad-

dresses interesting problem from the field of computational biology that is very topical as supported by a lot of recent literature. However, I have serious doubts about the aim of the work and I find it partially confusing. As such I cannot recommend it for publishing in the Nature Comm. Journal. My comments follow.

Thank you very much for the constructive criticism. We are happy to note that the reviewer finds this an interesting problem and topical. Thank you for pointing out the confusing parts. Below we address each of the reviewer's concerns in detail. Based on the reviewer's feedback, we have improved our manuscript in two ways: (i) We have done additional computations/simulations to clarify many of the valid questions the reviewer has raised. (ii) We have also rewritten some parts to clarify all points. Details are below:

First of all, I find confusing if the authors do understand the purpose of modelling, where a good model should provide predictions that can be tested experimentally. Here, the authors use experiments in a somewhat opposite way, to do predictions about their model, the intended use of which is not clear, as the model is not further applied to solve any specific experimental problem, so it looks like a half of the story is missing.

We would like to clarify this. We do not use experiments to make predictions about our model. We build our fine-grained model such that it reproduces nucleosome-level (scale $\sim 10 - 20$ nm) Micro-C experimental observations. Then we simulate the fine-grained model, using Brownian dynamics, and predict quantities at a much larger scale – at the scale of hundreds of nanometers to micrometers. Please note that many important physical experiments (light microscopy experiments, chromatin pulling experiments, etc.) measure chromatin properties at hundreds of nanometers to micrometers scale. So, in this work, we are predicting many experimentally measurable quantities at the scale of hundreds of nanometers to micrometers. We predict the radius of gyration of large coarse-grained chromatin beads, inter-bead 3D distances, spring constants, bending angle distributions, overlap, etc. Using the coarse-grained parameters, our model can simulate and predict chromatin behavior at a much larger length scale. All of our predictions can be tested using microscopy, chromatin pulling, and other biophysical experiments.

Importantly, we also show how these quantities depend on the coarse-graining scale. For example, we show that the spring constant at 1kb resolution differs from that at 10kb or 20 kb resolution. Similar scale-dependent results are presented for angle distribution, overlap, etc. The scale-dependence is also our prediction and can be tested experimentally. The scale-dependence is also important because even though everyone uses a polymer description of chromatin at different resolutions, until

now, we did not know how these physical quantities (size, spring constants, softness, etc.) depend on the resolution we choose. Here we are predicting precisely that.

The coarse-grained model and parameters we develop are further applied to solve a specific experimental problem of determining the radius of gyration and 3D distance. Note that the current coarse-grained models cannot accurately predict 3D distances and R_g without multiple fitting parameters such as bead size, spring constant, and strength of attractive interaction. We also have the advantage that we have quantified overlap in a scale-dependent manner, which is crucial for predicting the experimentally measured 3D distance. In the revised manuscript, we have clarified these; we have now written a subsection in the discussion on our predictions and potential experiments to test our predictions.

Also, in the systematic coarse-graining approach, some details of the structure are neglected for the sake of rationalizing the use of computer time and extending the physical time of simulations, perhaps, to a biological region of interest. The choice of the size of DNA portion to be coarse-grained is arbitrary but should be smaller than is the length scale of the studied problem.

It is true that the choice of the size of the DNA portion to be coarse-grained should be smaller than the length scale of the studied problem. In our work, we have presented results from 1 kb ($n_b = 5$, that is, five beads of 200bp) up to 100kb. Depending on the problem one is interested in studying, one can choose a suitable coarse-graining size. For example, in the last section, we have presented results with different coarse-graining sizes to study the Arsg gene locus. Instead of a gene, if one wants to simulate the whole chromatin or the whole set of chromosomes, one would choose a much larger coarse-graining size, like 10kb or 100kb. One of our interesting novel results here is how different polymer parameters, such as bead size, spring constant, bending angle distribution, overlap, etc., change as a function of the coarse-graining size. These results are not presented systematically in any published work so far, and they can be very useful as a reference for all future chromatin simulations. These are numbers predicted by our simulations, starting with a fine-grained model consistent with the Micro-C data.

In the model, we decided to make minimal assumptions. We assumed that different regions (of 200bp size) would be in contact according to the contact probability provided by Micro-C. There might be other chromatin details (like inter-nucleosome interaction potentials); all of those details are assumed to be leading to the contact probability we observe (which we ensure in our simulations).

In the revised manuscript, we have added a discussion on these aspects to clarify our arguments.

In Figure 1S, the authors show “Diameter (physical size) of simulation beads estimated for different chromatin segment lengths in various published papers”. I really don’t understand what the authors mean to say with this illustration, as they show the portions without any context. For example, the manuscript cites works by Mirny et al., who use various values for the size of the bead e.g. 440, 600 and recently 20 bp (Ref. 78), which were of course chosen with a particular purpose.

Our aim here was only to show that there is considerable variability in the numbers. We do not deny that this variability could arise from the appropriate context of the problem. In the revised manuscript, we have added more details to Figure S1.

However, we would like to also point out that our current work systematically examines the context of several gene regions. In the revised manuscript, we present results for ten different chromatin regions with different chromatin states and compare them to study how quantities vary along the genomic length. We also present the distributions of all quantities. We would like to stress that our study adds context to the observed variability from a nucleosome-level perspective (beyond broad hetero-/eu-chromatin states), giving us a novel understanding of how these numbers emerge. For example, we show how TAD boundaries or overlap/mixing lead to variability in numbers. Our results provide insight into these crucial aspects for understanding these numbers. Our results suggest that if one does not account for overlap/softness, the bead size will have a different interpretation.

In the manuscript’s abstract the authors argue about the importance of their computational findings, quote “unlike the prevalent notion, our findings argue that coarse-grained chromatin beads must be considered as soft particles that can overlap, and we propose an overlap parameter.” But again, what is the model aimed to be used for?

Our model aims to accurately predict 3D distance in chromatin simulations, as it accounts for scale-dependent properties like softness (overlap), spring constant, and angle distributions. This is crucial, as no previous work has quantified these physical parameters at the scale relevant for chromatin simulations, which range from 1kb to 100kb.

Specifically, if one were to simulate chromatin at a 10kb resolution, how much this 10kb chromatin would overlap is a prediction from our model. That would decide the softness/effective size of the bead. A work like ours at nucleosome resolution is essential for predicting this. Also, as mentioned earlier, the effective spring constant at 1kb coarse-graining is not the same as that at 10kb coarse-graining. This work, via systematic coarse-graining, shows how spring constants (and bead sizes,

bending angles, overlap parameters) vary as a function of coarse-graining size (Fig. 3 and Fig. 4 in the revised manuscript). These novel findings (and quantification) are highly relevant for anyone wanting to represent chromatin as a polymer (in particular, heterogeneous polymer) at the coarse-graining scale of their choice. In the last section, we use all of these to predict R_g or 3D distance.

Our findings have significant implications for accurately predicting 3D distance or R_g at the larger scale of TADs as well as in the scale of the whole chromatin. Accurate predictions of 3D distance are critical for predicting the interaction between different genes, enhancer activation, etc. Our model provides a more comprehensive and precise understanding of the physical parameters necessary for such predictions, which include bead size, overlap, and spring constant.

We have added a paragraph discussing the aim of the model in the Discussion and Conclusion section.

So, for example, the work by Liebermann-Aiden, cited in ref. 28, employed the hard-core potential and showed that the scaling of contacts with genomic distance that decay with exponent roughly equal to -1 corresponds to DNA organized as a fractal that is free of knots, that was very important and fascinating up to these days. Now, varying the softness of the beads would yield a different scaling exponent. So, what are the authors going to study with their model?

Thank you for this interesting question. The softness in itself will not change the scaling exponent. The key ingredient for the fractal globule scaling exponent is the idea that chromatin is a locally folded polymer, and far-away regions of the chromatin do not mix. That information is buried in the contact probability. If we ensure that our configurations satisfy the measured contact probability matrix, we will get the appropriate exponent. In the revised manuscript, we show that our coarse-grained beads do overlap (have softness), and at the same time, the contact probability as a function of genomic separation ($P(s)$ vs. s) from our simulation compares well with the experimental data (see Fig. R4 and Fig. S4).

Please note that our fine-grained model is built according to the Micro-C contact probability, and far-away regions of the polymer do not mix similar to the experimental system.

FIG. R4: The contact probability as a function of genomic distance obtained from Micro-C experiments is compared with our fine-grained simulation for the Ppm1g locus. The fact that they match reasonably well suggests that the simulated structure is close to what is observed experimentally.

In the revised manuscript, we use the iterative Boltzmann inversion method to compute the nature of non-bonded interactions explicitly. We indeed find that the coarse-grained chromatin beads interact with a soft interaction potential, and the softness increases with increasing coarse-graining. Given that the fine-grained model satisfies the contact probability scaling and the coarse-grained configurations obtained from the IBI method are consistent with the fine-grained model, we are confident that softness is crucial for modeling coarse-grained chromatin.

The idea of using the soft beads to model chromatin is not new, for example in the work by Giorgetti et al from 2014, cited as Ref. 10, the authors did a kind of Reverse Monte-Carlo simulations in N^2 parameter space, fitting also the softness of the beads on the experimental contact maps. But, Kadam et al should realize that the work by Giorgetti came out about a year after discovery of TAD's and it was a pioneering explanation of how TAD's are created. Since then, the "triangles" on contact maps had been becoming less fascinating as various works showed that they can be created by multiple ways, like protein linkers (works by Mario Nicodemi), supercoiling (works by Stasiak group), or even just a bond representing cohesin (works by Leonid Mirny, but also for instance j.cell.2015.11.024), but recent simulations by Mirny et al, shed light on how other prominent features observed on Micro-C are created, cited as ref. 78 in the manuscript.

We fully appreciate and acknowledge the large body of earlier modeling work mentioned by the reviewer above. We agree with the reviewer that the groups mentioned above have done pioneering

work, and they are very important. Hence we cite and discuss all of them in the revised manuscript. However, we would like to note that in the Giorgetti et al., 2014 work, they assume a square-well-like potential form for inter-bead interactions among 3kb beads. In our revised work, without assuming anything, we derive a functional form for the inter-bead soft potential using an iterative Boltzmann inversion method (see main manuscript Fig. 5). Moreover, we also quantify softness for different levels of coarse-graining, which can be used by anyone doing future simulations at all scales.

Our aim here is not to explain how TADs are created. One of our aims is, given the TAD structures and given the non-TAD (heterochromatin) structure (Fig. 2(c, d) of our manuscript), how physical quantities would vary along the genome and depending on whether one is studying a TAD interior or TAD boundary, at different scales. As mentioned earlier, we investigate how the coarse-graining scale is crucial for deciding the physical quantities like spring constant, bendability, bead size, and overlap. These points are not described in any of these papers. Either they assume or fit some of these parameters. That is, the above models start with a coarse-grained model assuming various physical parameters. Here we start with nucleosome resolution data enabling us to coarse-grain at multiple scales, compute spring constant, overlap, etc., and predict average 3D distances.

Our claim is: we have computed the relevant parameters for a polymer simulation of mESC chromatin. Anyone can choose whatever coarse-graining they want, take the corresponding parameters from our results, simulate coarse-grained chromatin satisfying contact probability, and predict average 3D distances reasonably well (within the region-to-region variability we show) without any fitting parameter. No one has made such a claim. In all the papers mentioned above, to predict 3D distance, one has to fit the bead size and assume spring constant, overlap (or lack of it), bendability, etc.

For all the reasons mentioned above, our work is novel. We have mentioned this now in the discussion of the revised manuscript.

The simulations in the manuscript test various models, in Model-II the authors use an explicit linker with nucleosomes. Conclusions from such the computational experiment are very tricky, as the authors use a simple beaded chain that does not have torsional stiffness, so the DNA at entry and exit points of the nucleosomes is unconstrained, and imposing just an angle would not work. In addition, the simulations of torsionally unconstrained chains possessing larger and smaller beads were already explored by Angelo Rosa (journal.pcbi.1004987) and showed no differences at larger scales.

Our aim of model-II is only to test if the R_g results obtained from model-I would still hold if we have

a different model with explicit linker length. We find that the R_g is approximately the same even if the model is different. This provides more confidence in our model-I results. Of course, model-II can be improved with many more details in the future. The absence of torsional stiffness is a limitation of model-II, and we have stated it in the revised manuscript. However, as mentioned, our aim is to test model-I with a different model, which is broadly sensible.

Thank you for pointing out the paper by Rosa and colleagues. We have cited and discussed the paper in the revised manuscript. Interestingly, the paper finds that below 100kb, the variability in thickness and flexibilities could affect the results. Since many of the enhancers and promoters can be within 100kb, this could be an interesting and biologically relevant point. In this work, we are predicting how various parameters vary along the genome, and this can be biologically relevant for enhancers/promoters below 100kb (e.g., it is known that in signaling systems, enhancers/promoters are very close to each other [7]. In some cases, they are ~ 10 kb apart. Very often, enhancers are less than 100kb apart [8]).

However, we would like to stress that we are predicting the magnitude itself (not just the variability) — we are predicting the magnitude of average bead size, spring constant, angles, etc. If the magnitude of the average changes, the results will change for all lengths (even above 100kb). Our other predictions related to softness/overlap will also be relevant at all lengths. We have discussed this in the revised manuscript.

All in all, I think the work by Kadam, S. et al. objects established coarse-graining approaches and polymer physics scaling laws as they say “prevalent notion”, but the authors show no use to it, the employment of soft beads is not new (Georgetti), simulations with explicit linker DNA miss torsional stiffness and are redundant to the existing works (Rosa), a great deal of important publications is missing (Nicodemi, Mirny, Stasiak, Plewczynski, Rosa), the authors seem to miss important points of the modelling approach and the authors just propose scaling principles for a model whose purpose is unclear. Given the number of issues, I cannot recommend the manuscript for publishing in the Nature Comm Journal. If there was anything aside my judgement to encourage the authors, I like the Figure 2c in the main text of the manuscript.

We have addressed each of the comments above in detail. However, we would like to summarise them in the context of the above comment: We predict the magnitude of the parameters like the bead size, spring constant, overlap, and angles and show how they depend on the scale of coarse-graining. No earlier work has pointed out how these parameters vary as we vary the coarse-graining

scale. All of these parameters are experimentally measurable. We also show that these parameters (overlap, size, etc.) are important for determining 3D distance and R_g . The purpose of the model is to predict the 3D distance and scale dependence of parameters. We thank the reviewer for pointing out missing references. We have cited them in the revised manuscript. About the “prevalent notion” of softness, we would like to make the following comment: Huge majority (maybe $\approx 99\%$) of the coarse-grained simulation papers use the Lennard-Jones potential for inter-bead interactions. The LJ potential quickly goes to infinity with negligible softness. The reason why a majority of papers still use LJ potential is that the prevalent notion in the community is that chromatin beads are closer to hard spheres. Our work provides systematic and quantitative estimates for softness. Moreover, we are also predicting a functional form for the inter-bead soft potential using an iterative Boltzmann inversion method (this is new data presented in the revised manuscript).

Thank you for appreciating our Fig. 2(c), where we show how parameters like bead size, angles, and overlap vary in a contact map-dependent manner. In the revised manuscript, we have presented a similar plot for a heterochromatic region, giving us insights about the physical properties depending on the genome architecture (see Fig. 2(d) in main manuscript and Fig. S6).

Response to reviewer 3:

In this work, Kadam et al. analyze the relevant properties of coarse-grained beads-on-a-spring models of chromatin, i.e., bead sizes, bond lengths and angles between neighboring beads, and corresponding stretching and bending stiffness. Using available nucleosome resolution micro-C contact data in mESC cells, they first derive ensembles of 3D structures reproducing the data by numerical calculations, exploiting the fact that the chromatin properties are mostly known at the nucleosome scale. Next, by systematically coarse-graining the obtained 3D models, they infer the scale dependence of such properties. Additionally, they show trends depending on the chromatin regions, i.e., euchromatin or heterochromatin, and on the genomic positions, e.g., at TADs boundaries compared to TAD interiors. Even though most of such trends are expected (e.g., that gyration radii, bond lengths and angles are larger at TAD boundaries) and in some cases previously observed, there is also some (little) interesting insight into chromatin structure. In particular, the bond angle distribution shows the presence of two peaks, one at the expected nucleosome bond angle and another at larger angles previously unobserved to my knowledge. Thus, I think the results are significant for an audience of experts working on chromatin polymer models to build increasingly reliable models and, to a lesser extent, for a broader audience since there is limited biological insight into chromatin structure. The methodology

is sound and well explained. However, some questions should be addressed to support the manuscript conclusions as detailed below.

Thank you very much for the positive comments on our manuscript. We have addressed the questions below in detail. Regarding the novelty/biological insight, we would like to mention that beyond computing novel quantities like angle distributions, quantifying softness, etc., our results on how the chromatin polymer parameters depend on the choice of coarse-graining scale are novel. That is, we show that chromatin properties at 1kb are different from chromatin properties at 10kb or 50kb. Quantification of this scale dependence is biologically relevant. So far, none of the papers show how spring constant, overlap (softness), angles, etc., depend on the scale you choose to study (coarse-graining scale).

Moreover, in the revised manuscript, we have derived a functional form for the inter-bead soft potential from the data. All our results are not only important for anyone doing chromatin simulations/analysis but also relevant for biologists trying to understand chromatin from a structure and function perspective. We show how chromatin properties vary depending on the epigenetic state as well as the contact map (TAD interior or boundary). To understand many biological processes (like enhancer activation, recombination, etc.), one has to predict 3D distance accurately. We show how overlap, chromatin flexibility, and heterogeneity are important in understanding 3D distance. Moreover, many of the biological processes like recombination, DNA breakage/repair, enhancer-activation, and spreading of histone modifications occur at the scale of nucleosomes. The 3D structure we predict at nucleosome resolution is crucial for understanding these functional aspects. Our work connects the coarse-grained picture (100 nm to μm scale experiments having a few kb or Mb resolution) with a nucleosome-resolution picture and will enable Hi-C or Microscopy experiments to extrapolate and predict nucleosome-level structure. This is highly relevant for understanding the biological functions that occur at nucleosome resolution. These novel results will interest not only the chromatin community but also the general audience interested in the coarse-graining of polymers/soft-matter/biological systems.

Major comments:

1. *The authors only model and perform analyses on two very small regions of chromatin, an 80 kb euchromatic region (Chr 5: 31.17-31.25 Mb) and a 160 kb heterochromatic region (Chr 7:70.24-70.4 Mb). Thus, my main concern is that all the obtained results, e.g., the precise quantitative differences found between eu- and heterochromatin, dependence of chromatin polymer properties on coarse-graining, etc. might depend on the specific chosen regions. If so, these results would not give help-*

ful insights for future simulations. The authors could repeat the analyses on other genomic regions (at least a few eu- and hetero- chromatic regions) to show that the results are consistent.

Thank you very much for this constructive comment. In the revised manuscript, we have simulated many more new regions—we have results for ten different genomic loci with a few euchromatic and a few heterochromatic regions. We now present R_g and other properties for all the regions. These new simulations also predict the variability/similarity among different gene regions. Details are discussed in the revised manuscript.

2. *One key finding is that coarse-grained beads (representing a certain genomic window) in polymer models must be considered as soft particles that can overlap with each other. However, some existing models in literature consider a coarse-grained bead of chromatin as a sequence of sub-beads, thus effectively taking into account the possibility of overlap, as, e.g., in Bianco et al. Nat. Gen. 2018, or in Chiariello et al. Cell Rep. 2020, where, e.g., a 4kb window is composed by a sequence of 16 non-overlapping sub-beads. I think the authors could comment that such models still allow for overlaps between coarse-grained chromatin windows and effectively lead to variable properties, e.g. gyration radii and bond lengths along chromatin.*

We agree with the reviewer that the models mentioned above (Bianco et al., Chiariello et al., etc.) do have sub-beads, and hence they will have the effect of the overlap. We have mentioned it in the revised manuscript.

However, the existing papers do not explicitly compute and quantify the overlap parameter (softness). In the revised manuscript, we have used an iterative Boltzmann inversion method and computed an effective inter-bead potential from the fine-grained model data (see Fig. R3 and Fig. 5). The potential clearly shows softness, and the softness is quantified. Moreover, the existing studies do not perform systematic coarse-graining at different scales to predict scale-dependent properties. The novel steps we have taken to go to the next level are: (i) we are making use of nucleosome-resolution contact map data, accounting for the interaction of each bead. (ii) We are systematically coarse-graining and computing quantities like spring constant, bending angles, and overlap explicitly for different levels of coarse-graining. Our results are extremely useful if any other group wants to simulate or describe chromatin organization at any coarse-grained level of their choice. In particular, to describe chromatin at many Mbp or the whole genome scale (Gbp-sized genomes), our results are essential. Our results are also helpful for experimentalists to describe chromatin configurations based on Hi-C data at different resolutions so that accurate 3D distances can be recovered. Moreover, our nucleosome-resolution configurations can also be used to understand biological phenomena at

this resolution, like recombination, DNA breakage/repair, or spreading of histone modifications (these events happen at \sim nucleosome solution).

3. *The similarity between model and micro-C contact data is measured by Pearson correlation, however it is recognized that it is not a good metric since dominated by the decaying trend of contact probability with the genomic distance. The authors could consider different metrics, e.g., the HiCRep stratum adjusted correlation coefficient [Yang et al. Genome Res 2017], or the Pearson correlation as a function of the genomic distance, or others.*

Thank you for the suggestion. In the revised manuscript, we have computed the stratum-adjusted correlation coefficient (SCC) for all the regions we simulated (see Table S1). Most of the regions show SCC values higher than 0.9. We have also presented contact probability as a function of genomic distance from our simulation and compared it with the corresponding experimental data. Experimentally obtained contact probability from Micro-C as a function of genomic distance matches well with the corresponding contact probability from our simulations (see Fig. S4).

4. *The authors show that R_g , l_{cg} , θ and overlap values have peaks at TAD boundaries for the euchromatin region (Figure 2c). I wonder if some pattern can also be found for the heterochromatic region (e.g. are the peaks of R_g in the same locations as peaks of l_{cg} , and are they located in specific regions as TAD boundaries?), even though here there is less evident TAD like structure.*

Thank you for this suggestion. In the revised manuscript, we are presenting this data for the heterochromatic regions as well (Fig. R5; also see Fig. 2(d) in main manuscript and Fig. S6). Here we observe that the R_g , l_{cg} , and θ_{cg} have higher values at the boundary of heterochromatic domains, while the overlap has a smaller value at these boundary locations.

FIG. R5: The R_g , l_{cg} , θ_{cg} , and overlap parameter is plotted as a function of genomic location for a heterochromatic region (Gm29683 locus). Here we observe different behavior at the boundary of heterochromatic domains.

5. *The Conclusions could be expanded to better summarize the main findings.*

Thank you for this suggestion. In the revised manuscript, we have expanded the conclusion and have improved the summary of our findings.

6. *It would be useful to add a few more details about the performed polymer simulations in the Supplementary Methods, e.g., what are the initial polymer configurations, how long are the simulations, how many timesteps until the equilibrium is reached.*

Thank you for this suggestion. In the revised manuscript, we have added the details regarding the polymer simulations (see SI Sec. S1.A).

Minor comments:

1. *I think the acronym FG for fine-grained is not ideal since it can be confused with the same used for Fractal Globule model of chromatin*

In the revised manuscript, we have gotten rid of this acronym and are using the full phrase “Fine-Grained” since it is only used a few times.

2. In Figure 2D caption I would refer to the experimental data as “orange squares” instead of “data points” to avoid confusion. Additionally, the authors could clarify the type of data in the main text (or also in the caption) because the fact that they are relative to *Drosophila* (and to a repressed region?) only appears in the plot.

We have updated the caption in the revised manuscript to incorporate these changes.

3. I wonder why the plots in Figure 3 (g) and (h) do not start with $n_b = 1$.

Our minimum level coarse-graining was $n_b = 5$; now, we have computed the spring constant values below $n_b = 5$ as well. In the revised manuscript, we have updated the figures to show the spring constant values below $n_b = 5$ (Fig. R6 and Fig. 3).

FIG. R6: The spring constant as a function of coarse-graining level (n_b) for various genomic locations in two different units.

4. In Supplementary Methods, first section, “we set $C_{ij} = 1$ if $P_{ij}^{pr} < r_n$,” < should be >

In the revised manuscript, we have corrected the typo.

-
- [1] T.-H. S. Hsieh, C. Cattoglio, E. Slobodyanyuk, A. S. Hansen, O. J. Rando, R. Tjian, and X. Darzacq, *Molecular cell* **78**, 539 (2020).
- [2] W. Lu, J. N. Onuchic, and M. Di Pierro, *PLOS Computational Biology* **19**, e1011013 (2023).
- [3] A. Buckle, C. A. Brackley, S. Boyle, D. Marenduzzo, and N. Gilbert, *Molecular cell* **72**, 786 (2018).
- [4] H. A. Shaban, R. Barth, L. Recoules, and K. Bystricky, *Genome biology* **21**, 1 (2020).
- [5] G. Spracklin, N. Abdennur, M. Imakaev, N. Chowdhury, S. Pradhan, L. A. Mirny, and J. Dekker, *Nature Structural & Molecular Biology* pp. 1–14 (2022).
- [6] E. M. Hildebrand, K. Polovnikov, B. Dekker, Y. Liu, D. L. Lafontaine, A. N. Fox, Y. Li, S. V. Venev, L. A. Mirny, and J. Dekker, *bioRxiv* pp. 2022–10 (2022).
- [7] S. Oh, J. Shao, J. Mitra, F. Xiong, M. D’Antonio, R. Wang, I. Garcia-Bassets, Q. Ma, X. Zhu, J.-H. Lee, et al., *Nature* **595**, 735 (2021).
- [8] Z. Islam, B. Saravanan, K. Walavalkar, U. Farooq, A. K. Singh, S. Radhakrishnan, J. Thakur, A. Pandit, S. Henikoff, and D. Notani, *Genome Research* **33**, 1 (2023).

REVIEWERS' COMMENTS

Reviewer #1 (Remarks to the Author):

I appreciate the authors' efforts in addressing the concerns I raised during the review process. Upon reviewing the revised manuscript, I am satisfied that all of my questions have been thoroughly answered and that significant improvements have been made. As a result, I recommend the publication of the manuscript.

Reviewer #2 (Remarks to the Author):

The manuscript by Kadam et al entitled "Predicting scale-dependent chromatin polymer properties from systematic coarse-graining" addresses very topical problem from computational biology and research of DNA. The research of DNA marked several milestones, from discovery of inheritance rules, nucleins, double helix, sequencing of the whole genome and the recent research focuses on revealing relations between chromosomal DNA structure and biological functions. The structure of the chromosomal DNA is revealed thanks to development of new group of methods called chromosome conformation capture. This method works with large amounts of data and provides only 2D interpretation of the 3D chromosomal structure. This is where computer modelling and molecular simulations became indispensable method since the 3D information on the chromosomal DNA cannot be currently obtained by any other way. An important approach in molecular simulations are coarse-grained models. These models neglect some structural details by replacing them by beads, the coarse-grains, what allows saved computational resources to be used for studying systems on longer length and time scales. Development of such models over the past 40 years for studying DNA was made possible based on the assumption that DNA is a polymer represented by a chain of beads and also based on discoveries in polymer physics, scaling laws and theories. The bead represents a coarse-grained portion of DNA while the size of the portion is free to choose and there exists plentiful of works using beads that represent variety of portions of DNA. In context with the advancement of chromosome conformation capture methods and improving resolution, current trend is using smaller coarse-grained portions of the DNA.

The work by Kadam et al is a contribution to this developing field of DNA modelling. The work however pursues another direction developing a systematic coarse-graining approach towards using beads that represent very large portions of DNA. The authors discover, that in the case of coarse-graining of larger portions of DNA, the beads should be modelled by soft potentials. The authors check their simulations with the contact maps obtained by chromosome conformation capture. It is known though, that polymer models with variable bond lengths and beads able to overlap can perfectly reproduce the experimental contact maps. However, when using beads representing larger portions of DNA the simulations can hardly reveal what happens on structural level of DNA and how DNA is organized in space. For example, such a region of current interest in structural organization of DNA is represented by "bubs" of contacts, that apparently exists on contact maps, but a soft bead model will coarse-grain over this information and represent them as unphysically overlapped beads. The authors explain the intended use of the model to predict properties like gyration radius (R_g) on the length scales of the whole chromosomes that can be experimentally measured by photo-labelling microscopy. Hence, I wonder if the model just does not serve a self-purpose, as I believe the data of R_g from microscopy and light scattering are already there. So why would someone use relatively expensive computational methods in terms of skilled bioinformaticians and high-performance computing infrastructure to "predict" something what is already available?

The authors admit, that soft potential may lead to passages and enhance knotting. They do not appear to be worried about it and support their statement by discussing a very good and recent work by Hildebrand et al (Ref. 98 in the main manuscript) and saying "topoisomerase activity is required to remove these crossings and have entanglement-free interphase chromosomes". They further conclude

that "This implies that more accurate dynamics would require the presence of enzymes like topoisomerase that actively regulate chromosome topology in terms of entanglements." I believe, the reference to the work by Hildebrand et al is badly misinterpreted and even harmful for current understanding of DNA organization in the context of the work by Kadam et al. Hildebrand et al in their work correctly state that "DNA in interphase chromosomes exists in a form of unknotted fractal globule, topoisomerases are involved in the process to reach this state, but it remains a mystery how they do it." Hildebrand et al then propose a mechanism, how topoisomerases could work to remove entanglements in a specific case, when chromosomal DNA undergoes transition from mitotic to interphase chromosomes. The data in the work by Kadam et al are obtained on interphase chromosomes, so they should be already in the topological state of unknotted fractal globule. In the other part of manuscript, the authors show comparison of the decay of contacts obtained by Inverse Boltzmann Iterative method (IBI). The method fits density of energy states involving also stochastic step that does not restrict passages, hence in the model implementing soft beads the resulting 3D structures can be more entangled. It is known that topological state of the DNA affects its biological functions and biophysical and physical properties of the DNA, including elasticity mentioned in the manuscript. I suggest the authors to check the topological state of the resulting structures by available topological software prior to discussing applicability of their model.

The idea of using soft beads to model larger portions of DNA might be driven by a bias of the model originating from the use of spherical geometry of the beads and increasing dissimilarity of the polymer chain when replacing larger and larger portions of DNA by spherical beads. The coarse graining approach is not limited to using spherical beads, and in the past, beads with various geometry like cylinders, tubes, ellipsoids, slabs, etc were used in the coarse-grained approach to model DNA. The soft potential seems related to de Gennes blob theory that shows different regimes of scaling inside blobs and between blobs. But, it is hard to say, as the manuscript does not draw links of the soft bead model to the context with existing polymer physics scaling laws, theories and models, such as Kremer-Grest model currently having around 4000 citations. For this reason, the manuscript is very difficult to read and raises questions. Aside of the questions a reader may have, there are eleven (11) questions in the manuscript raised by the authors themselves. I believe though, some of the questions raised by authors can be answered by applying some fundamental polymer physics that is heavily omitted in a manuscript that is focused on development of a new coarse-grained modelling approach to DNA.

In conclusion, I would say that the work by Kadam et al focuses very important field of molecular simulations of DNA and development of new models, but in order to truly promote the molecular simulations as indispensable method the work should demonstrate the use of the model in a joint experimental work or demonstrate its advantage by thorough discussion with existing polymer physics theory and models. Given the number of questions raised by authors themselves, that I believe can be answered by applying the existing polymer theoretical physics and misinterpretations of existing works, I cannot recommend the manuscript in its current form for publishing in the Nature Comm. Journal.

Reviewer #3 (Remarks to the Author):

My previous concerns have been addressed in the revised manuscript, which has been highly improved by including several novel simulations of different chromatin loci and novel analyses. I also appreciate the effort to stress the novelty of the work, which is now more evident to me. Therefore, I think the manuscript is suitable for publication in Nature Communications, after addressing the following minor comments remaining:

- I think there is some mismatch in the captions of figures FIG. S7 and FIG. S8 as they do not correspond to the figure content.

- Since the gyration radius is different for different beads, what is the R_g in the definition of overlap precisely? Is it the average R_g of the two adjacent beads? Please clarify.